Methods

# Evaluation of genetic demultiplexing of single-cell sequencing data from model species

Joseph F Cardiello[1], Alberto Joven Araus[2] , Sarantis Giatrellis[2], Clement Helsens[3] , András Simon[2], Nicholas D Leigh[1]

**Single-cell sequencing (sc-seq) provides a species agnostic tool to study cellular processes. However, these technologies are expensive and require sufficient cell quantities and biological replicates to avoid artifactual results. An option to address these problems is pooling cells from multiple individuals into one sc-seq library. In humans, genotype-based computational separation (i.e., demultiplexing) of pooled sc-seq samples is common. This approach would be instrumental for studying non-isogenic model organisms. We set out to determine whether genotype-based demultiplexing could be more broadly applied among species ranging from zebrafish to non-human primates. Using such non-isogenic species, we benchmark genotype-based demultiplexing of pooled sc-seq datasets against various ground truths. We demonstrate that genotype-based demultiplexing of pooled sc-seq samples can be used with confidence in several non-isogenic model organisms and uncover limitations of this method. Importantly, the only genomic resource required for this approach is sc-seq data and a de novo transcriptome. The incorporation of pooling into sc-seq study designs will decrease cost while simultaneously increasing the reproducibility and experimental options in non-isogenic model organisms.**

## Introduction

Over the last decade, single-cell RNA sequencing (scRNA-seq) has exploded in popularity as a species agnostic tool for studying gene expression at the level of individual cells (Klein et al, 2015; Macosko et al, 2015; Villani et al, 2017; Han et al, 2018). The biggest impact has been on species in which study at the cellular level was long difficult or impossible because of lack of species-specific antibodies or other reagents (i.e., all species other than rodents and primates). Although extremely powerful, the scRNA-seq approaches that have risen to prominence are expensive and low throughput for biological replicates. This is a serious drawback as the lack of biological replicates has also been shown to be a major cause of

false discoveries (Hicks et al, 2018; Squair et al, 2021). To limit artifacts in scRNA-seq, there is a critical need for approaches that allow for adequately powered experiments (Squair et al, 2021; Zimmerman et al, 2021). Sample pooling is an effective means to increase biological replicate throughput while simultaneously decreasing batch effects and costs. Sample pooling can also enable scRNA-seq experiments with limited cell quantities and can identify doublets which, depending on cell loading, can make up a substantial portion of sequenced droplets (DePasquale et al, 2019).

In fields working with low cell numbers, like developmental biology, pooling of samples from multiple animals with no sample labeling method, or intention of demultiplexing has become a standard practice. This approach lacks advantages of true replicates because there is no way to assess the data for representation of all samples or variation between samples. The inability to demux pooled samples thus lacks the ability to account for replicate variation and perform replicate strengthened differential expression analysis (Squair et al, 2021; Zimmerman et al, 2021). Because of these benefits of biological replicates, pooled samples in which biological replicates were collected would ideally be able to be demultiplexed, providing information on the origin of each cell in the experiment.

Methods for analyzing pooled data and for enabling the demultiplexing (also known as demuxing) of pooled scRNA-seq samples are varied in concept and accuracy and have been recently reviewed (Zhang et al, 2022). To preserve the benefits of true biological replicates within pooled samples, experimental protocols for demultiplexing of pooled scRNA-seq samples have been developed. These methods include pooling cells from transgenic animals expressing a distinct transgene (Lin et al, 2021) or oligonucleotide (Shin et al, 2019). The most common method for scRNA-seq pooling is cell hashing, in which cells from each sample are labeled with antibodies (Stoeckius et al, 2018), lipids (McGinnis et al, 2019), or chemicals (Gehring et al, 2020) tethered to an oligonucleotide label that links gene expression data from each cell to the cellular origin (i.e., cell multiplexing oligonucleotide (CMO) label). A downside to these label-based approaches is that they each have varying degrees of efficiency, require sufficient cell numbers, have costs associated with their application, are not compatible with all species, and have a chance of failing (Zhang et al, 2022).

[1]Molecular Medicine and Gene Therapy, Wallenberg Centre for Molecular Medicine, Lund Stem Cell Center, Lund University, Lund, Sweden   [2]Department of Cell and Molecular Biology, Karolinska Institute, Stockholm, Sweden   [3]Institute of Bioengineering, Ecole Polytechnique Fédérale de Lausanne, Lausanne, Switzerland

Correspondence: nicholas.leigh@med.lu.se

   

In contrast to these experimental demultiplexing approaches, computational methods have been developed to demultiplex pooled human samples without any labeling regimen using the natural genetic differences between individuals. These approaches detect genetic differences between samples at sites of single-nucleotide polymorphisms (SNPs) and implement demultiplexing based on differential distributions of these SNPs between samples. SNP-based approaches have been benchmarked and shown to be highly effective at separating human samples (Kang et al, 2018; Huang et al, 2019; Xu et al, 2019; Heaton et al, 2020; Weber et al, 2021). Relative to laboratory species, SNP-based computational approaches are expected to perform well in human samples because of their high genetic diversity and wealth of available genomic resources. Outside of human samples, SNP-based demultiplexing has been applied to demultiplex other species: plasmodium samples, and across mouse strains (Heaton et al, 2020; Mylka et al, 2022). Although these results are promising, thorough evaluation of the strengths, weaknesses, and limitations of SNP demuxers in model organisms—through comparison of results to ground truths derived from orthogonal wet laboratory–based methods—is required to demonstrate whether this is an approach that warrants widespread use.

In this project, we set out to learn whether SNP-based demultiplexers work in an array of non-human species. We benchmarked available SNP-based demultiplexing programs and found that most are highly accurate on model organism datasets. We then selected the demultiplexing tool with the broadest potential usability, souporcell (Heaton et al, 2020), to more thoroughly test against a diverse range of species and wet laboratory–based multiplexing methods. We found that SNP-based demuxers can successfully demultiplex single-nuclei RNA-seq and scRNA-seq datasets without any prior information about the samples and with only a de novo transcriptome as a reference. This suggests that SNP-based demuxers can facilitate effective experimental design via the demultiplexing of pooled scRNA-seq data from a vast range of species. This lowers burdens and substantially broadens the use of scRNA-seq to study cellular processes in most organisms.

## Results

### Highly accurate SNP–based demultiplexing of in silico pooled sc-seq from zebrafish, monkey, and axolotl

We first explored the performance of SNP-based demultiplexing methods when applied to a published zebrafish dataset. Two useful resources for applying SNP-based demultiplexing are available for zebrafish, a high-quality genome (Howe et al, 2013), and a common SNP variant, variant call format (VCF) file for the species (LaFave et al, 2014). A recent study collected single animal scRNA-seq datasets from the thymus of zebrafish (Rubin et al, 2022). These data present an opportunity to synthetically pool samples to test SNP-based demultiplexing on data from a non-human species. Although synthetically pooled data are less challenging for an SNP-based demultiplexing algorithm than experimentally pooled

samples, the individually sequenced libraries provide a ground truth for definitive benchmarking of SNP-based assignments.

After processing each zebrafish scRNA-seq sample individually, we performed in silico pooling of three samples (Fig 1A). This synthetic pooling creates a pooled sample that has cell origin information and synthetic doublets. The production of synthetic doublets bolsters the ability to test and validate the accuracy of cell and doublet assignments. The synthetically pooled data were then demultiplexed with the SNP-based demuxers souporcell (Heaton et al, 2020), Vireo (Huang et al, 2019), Freemuxlet (https://github.com/statgen/popscle; Kang et al, 2018), and scSplit (Xu et al, 2019)—all of which do not require prior genotyping information. We found that souporcell, Vireo, and Freemuxlet gave highly concordant results, with scSplit giving divergent assignments for one individual (Fig S1A and B). The computational requirements for this dataset were similar between these tools, with Freemuxlet finishing the quickest and scSplit requiring the least memory (Table S1). We also ran souporcell on this dataset with and without providing the common SNPs VCF file. We found that for this zebrafish dataset, the use of the common VCF file had minimal overall effect on the souporcell demultiplexing results, with less than 1% of cell assignments differing (11/1,312 cells). Given the highly similar results and compute requirements from souporcell, Vireo, and Freemuxlet, we chose to move forward with more thorough analysis of souporcell because this tool has the broadest universal applicability for samples from traditional and non-traditional model species due to its lesser input requirements (Fig S2).

To further investigate the demultiplexing accuracy of souporcell, SNP-based cell assignments were assessed for correlations with ground truth animal origin. We found a strong agreement between souporcell assignments and ground truth animal identities (Fig 1B and C). We then analyzed how many cells of each known animal origin were assigned to each animal by souporcell (Fig 1C, left as total cell quantities, right as percentages). SNP-based demultiplexing of this synthetically pooled zebrafish scRNA-seq data was highly accurate, resulting in correct cell assignments of 99–100% of cells based on their known animal origin (Fig 1C). This suggests that genetic demultiplexing is a viable means to enable sample pooling and subsequent demultiplexing in zebrafish.

The presence of doublets in single-cell RNA sequencing is a major confounder. A doublet is a droplet represented by a single-cell barcode that contains more than one cell. In these in silico pools, true doublets can be identified with absolute certainty because we have origin information. When comparing SNP-based demultiplexing results to ground truth, "confirmed doublets" are cells that were assigned doublets by both the ground truth and demuxer. Furthermore, "contested doublet" refers to cells in which the experimentally derived ground truth and SNP-based demuxer result disagree about a potential doublet. We thus investigated the doublet detection capacity of souporcell for heterotypic true doublets (i.e., doublets from two genetically distinct individuals in a defined in silico pool). Homotypic true doublets created during synthetic pooling were removed, as souporcell relies on intergenotypic doublet detection. We found that souporcell missed almost half of the synthetic heterotypic true doublets in the pooled dataset (Fig 1C). The relatively poor performance (55% heterotypic true doublets identified) of souporcell at identifying synthetic heterotypic doublets from this zebrafish

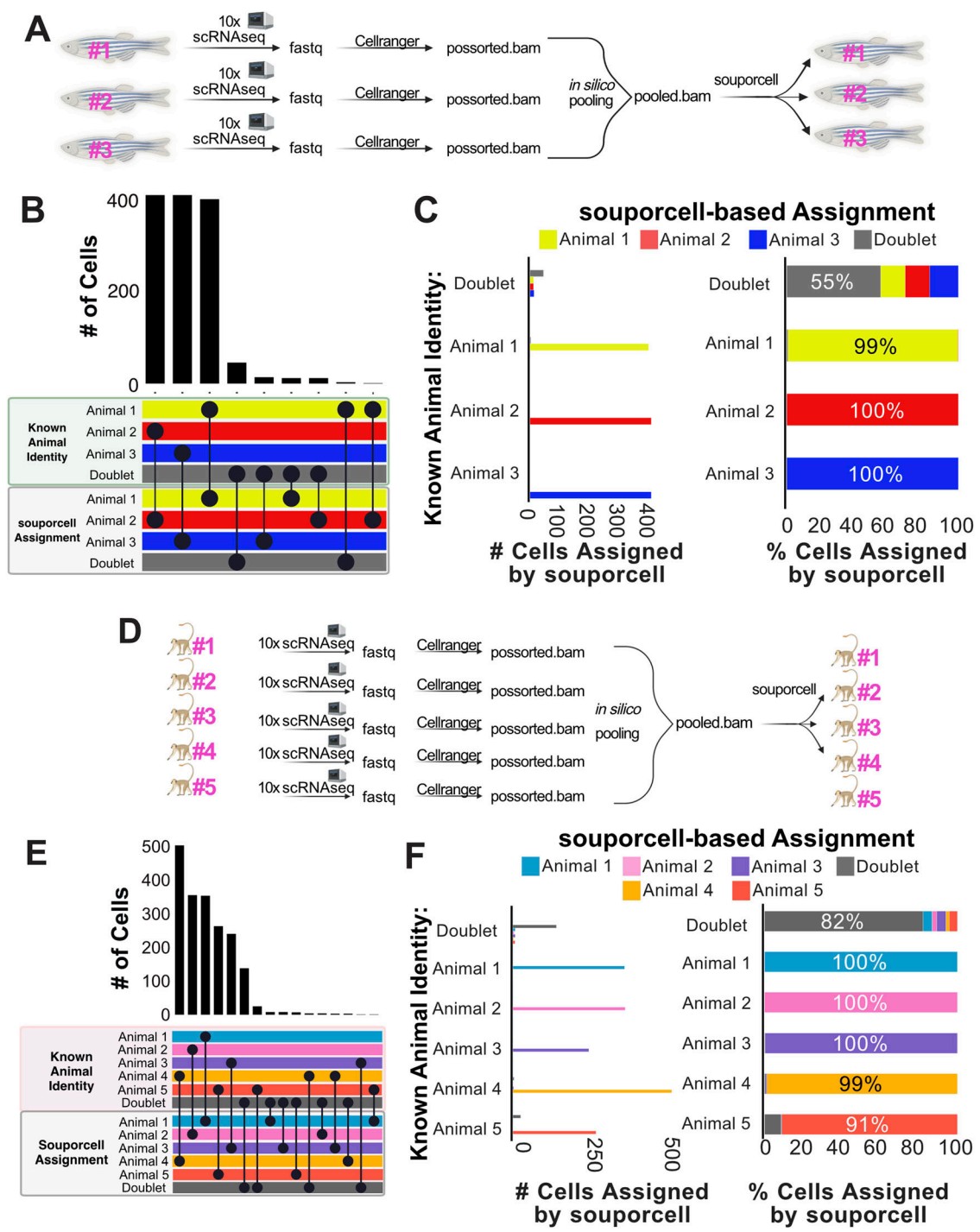

**Figure 1. SNP-based demultiplexing enables demultiplexing of synthetically pooled zebrafish and African green monkey scRNA-seq data.**
**(A)** Conceptual diagram of benchmarking analysis for zebrafish data (n = 3). **(B)** Upset plot comparing cell assignments by souporcell to known animal identities for each cell. Souporcell assignments were matched with known identities by correlation analysis. **(C)** Bar plots quantifying the distribution of souporcell assignments for cells from each animal. Left: of cells known to originate from each animal, the number of those cells assigned by souporcell to each animal is plotted. Right: of cells known to originate from each animal, souporcell assignments are shown as a percentage of total cells assigned to that animal. **(D, E, F)** Same as (A, B, C) but for African green monkey (n = 5) scRNA-seq data.

dataset made doublet detection the largest discrepancy in this analysis. Souporcell was not alone in its poor doublet detection performance, with Freemuxlet and Vireo misassigning many of the same doublets as

souporcell (Fig S1B). This suggests that SNP-demultiplexing tools struggle with heterotypic doublet detection when only three animals are pooled.

We next decided to in silico benchmark these SNP demultiplexers on another potentially inbred population, the African green monkey. The African green monkey is a pre-clinically relevant species, with a published genome (Warren et al, 2015) and SNP variant VCF file available for use (Huang et al, 2015). For this experiment, five individually sequenced green monkey scRNA-seq datasets were filtered for high-quality cells, pooled in silico, and subsequently demultiplexed (Fig 1D) (Speranza et al, 2021). Our results from this demultiplexing closely mirrored that of the zebrafish dataset in terms of accuracy and the similarity of results from independent SNP-based demultiplexing programs (Fig S3). Compared with demultiplexing of three zebrafish samples, the doublet detection observed with this pool of five monkey samples was improved (82% versus 55%) (Fig 1E and F). We thus tested whether the number of samples pooled contributes to doublet detection efficiency. We pooled three of the same monkey samples and SNP demultiplexed them using souporcell. We found that souporcell only called 66% of true heterotypic doublets from this pool of three monkey samples (Fig S4). This suggests that the number of samples in a pool contributes to efficiency of heterotypic doublet detection. Overall, these results indicate that SNP-based demultiplexing can be a highly accurate and efficient method for demultiplexing pooled single-cell data from non-human primates.

Finally, we also assessed results of SNP-based demultiplexing of synthetically pooled single-nuclei data from axolotl. The axolotl is an example of the type of organism for which scRNA-seq has enabled cell level study of regeneration and immunology for the first time (Gerber et al, 2018; Leigh et al, 2018; Rodgers et al, 2020; Lin et al, 2021; Lust et al, 2022; Ye et al, 2022). A genome (Nowoshilow et al, 2018; Smith et al, 2019) and SNP variant VCF file (Timoshevskaya et al, 2021) are available for the axolotl, but its large genome provides a distinct challenge when using computational tools. We used in silico pooling for three individually sequenced axolotl single-nuclei RNA-seq datasets (Fig S5A) (Lust et al, 2022). 95–99% of the cells from each individual axolotl dataset were correctly demultiplexed from the pool (Fig S5B and C). Similar to the three sample zebrafish datasets, we found that souporcell failed to identify a large proportion of the synthetic heterotypic doublets (Fig S5C). These results suggest that SNP-based demultiplexing may accurately demultiplex pooled samples in any single-cell compatible organism with genetic heterogeneity.

### Identifying the limits of SNP-based demultiplexing: inbreeding and SNPs density

With the promising results observed in zebrafish, African green monkey, and axolotl, we next investigated the minimum level of animal genetic variation required to enable SNP-based demultiplexing. To do this, we used previously published data from highly inbred isogenic mice (Crowl et al, 2022). Our efforts to apply SNP-based demultiplexing to an in silico–pooled mixture of three C57BL/6 mice samples (Crowl et al, 2022) were unsuccessful, with 99.7% of cells being unassigned by souporcell or Vireo (data not shown). C57BL/6 mice are nearly genetically identical with only ~15,000 SNPs and insertion–deletion mutations (indels) (Doran et al, 2016), providing evidence that SNP-based demuxers are unable to separate biological replicates from highly inbred mice. We also attempted SNP-based demultiplexing of in silico–pooled data from DBA/1J mice (Shinozaki et al, 2022) which have ~5 × 10^6 total SNPs and indels (Doran et al, 2016). Again, 99.7% of cells were unable to be assigned via souporcell (data not shown), suggesting that SNP-based demultiplexing is incompatible with highly inbred animals. These findings are in line with other claims that SNP-based demultiplexers are unable to separate samples within the same mouse strain (Mylka et al, 2022). Although the vast majority of murine work is performed on a single strain, pooling different strains reportedly does allow for SNP-based demultiplexing (Mylka et al, 2022).

To try to determine SNP frequency thresholds required for successful SNP-based demultiplexing, we investigated the density of SNPs in all the datasets used thus far. We found that both inbred mouse strains had less than 0.2 SNPs/kilobase (kb) (0.013 and 0.19 for C57BL/6 and DBA1/J, respectively). In comparison to the inbred mouse strains, the SNP density of the successfully demultiplexed datasets was higher at 0.34/kb (axolotl), 0.86/kb (African green monkey), and 3.57/kb (zebrafish). Although an exact quantitative analysis of this possible genetic cutoff would be useful, these results imply that a range of 0.2–0.34 SNPs/kb may be the minimum required within a sc-seq dataset for SNP-based demultiplexing.

### SNP-based demultiplexing of high sample number pools

Our previous results with in silico–pooled scRNA-seq data indicated high accuracy of SNP-based demultiplexers, but this setup lacks challenges of physically pooled scRNA-seq data, like ambient RNA and real heterotypic doublets. Therefore, we were interested in benchmarking SNP-based demultiplexing accuracy in a realistic and challenging experimental scenario: experimentally pooled cells from siblings, with a high sample number and without a common SNPs VCF file. We analyzed a published dataset of *Xenopus laevis* scRNA-seq data containing eight experimentally pooled samples from three *Xenopus* transgenic lines that each overexpress a different fluorescent gene (Lin et al, 2021) (Fig 2A). *Xenopus* is another common laboratory animal that has a published high-quality genome (Session et al, 2016).

To identify the transgenic line origin based on fluorescent mRNA counts, we co-opted MULTIseqDemux (McGinnis et al, 2019) to assign donor identities based on the transgenic line–expressed fluorescent gene counts. Although this approach succeeded in assigning the *Xenopus* transgenic line of origin, the low number of fluorescent gene counts left many cells without sufficient data to make an assignment prediction (Figs 2B and S6). This low detection of fluorescent gene transcripts is a common problem when using fluorescent marker genes for demuxing pooled data (Lin et al, 2021) and underscores the importance of alternative means to demultiplex pooled data. To avoid benchmarking results with low-quality cells, stringent filtering was used to ensure that only cells likely to have accurate origin assignment by the fluorescent gene–based demultiplexing method would be used to benchmark SNP-based demultiplexing.

We first assessed the *Xenopus* data for correlation between fluorescent and SNP-based cell assignments. We observed a remarkable similarity in demultiplexing the eight-animal dataset with these two orthologous methods (Fig 2C–E). To further quantify the

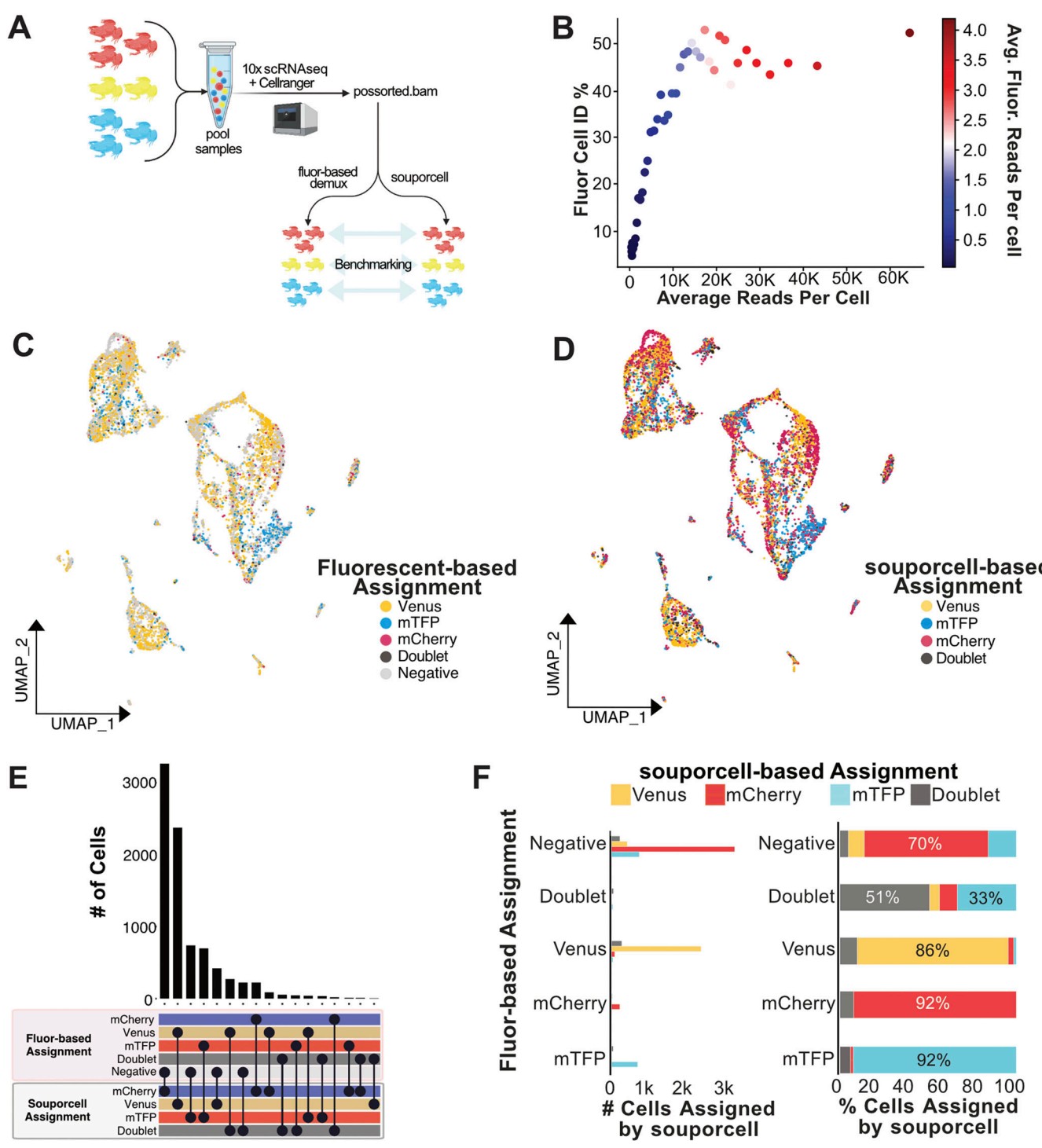

**Figure 2. Experimentally pooled, high sample number *Xenopus* data demultiplexed by SNP-based methods.**
**(A)** Conceptual diagram of benchmarking analysis for *Xenopus* data (n = 8). **(B)** Cell identification percentage (Fluor Cell ID %) by fluorescent-based assignment is plotted against average read depth. All cells were sorted by read depth and binned into 40 groups before calculating Fluor Cell ID%, and average total, and fluorescent read depth. Fluor Cell ID % is defined as the percentage of cells in each bin that were assigned to any of the three transgenic animal identities by fluorescent-based demultiplexing analysis. Binned data are colored by the average number of summed fluorescent reads per cell. Subsequent analysis plots focused on cells with between 5,000 and 40,000 mapped reads, and >0 summed fluorescent gene reads. **(C)** UMAP plot of *Xenopus*-pooled scRNA-seq data colored by fluorescent-based cell assignments. **(D)** UMAP plot of *Xenopus*-pooled scRNA-seq data colored by souporcell assignments relabelled according to correlating transgenic animal name. **(E)** Upset plot comparing cell assignments by souporcell to fluorescent based assignments. **(F)** Bar plots quantifying the distribution of souporcell assignments for cells from each animal. Left: of cells assigned to each transgenic line through fluorescent-based assignments, the number of those cells assigned by souporcell to each category is plotted. Right: of cells assigned to each transgenic line through fluorescent-based assignments, souporcell assignments are shown as a percentage of total cells in that category.

souporcell assignment accuracy, we evaluated how souporcell performed in comparison to transgenic line assignments made by fluorescent-based assignment (Fig 2F). We found high agreement between the two methods, as demonstrated by the souporcell assignment agreement on 86–92% of cells in each sample (Fig 2F). Furthermore, many cells were assigned as "negative" by the fluorescent-based approach, which could then be computationally rescued and assigned a transgenic line by souporcell. This is particularly notable for the mCherry-expressing cells, which were difficult to assign by the fluorescent-based assignment method (Fig 2C and D). Also of interest, we found that one of the eight transgenic animals was almost completely absent from the dataset, only displaying dozens of total cells. Although individual animal assignments by souporcell could not be validated, this suggests animal dropout in this dataset that would not be identified without demultiplexing.

As with the in silico experiment, doublet assignments were the biggest discrepancy. It is not clear from the available data whether cells are true doublets being identified only by souporcell, or if the SNP-based demuxer is over-assigning doublets. However, assuming the worst-case scenario: that souporcell is over-assigning doublets, this is a relatively harmless mistake considering that a standard analysis would subsequently remove these doublets from downstream analysis. Overall, these results display a high degree of accuracy for SNP-based assignments in a complex, experimentally pooled mixture of *Xenopus* cells.

Next, we sought to evaluate whether souporcell could produce assignments in a previously published dataset of a pool of 30 zebrafish embryos (Metikala et al, 2021). We found this dataset to contain 25,279 cells, which according to the manufacturer would result in ~20% of droplets being doublets. This would be considered a superloaded sample as it is over the supported cell recovery recommendations. However, this type of experiment is common and would be benefit from SNP-based demultiplexing. We applied souporcell to this dataset and found that it gave assignments for 95.1% of cells (24,047 of 25,279). Although this experiment does not have a ground truth to compare with, we found that the doublet rate reported by souporcell was 18.6%, close to the expected 20%. All 30 embryos were identified in this final dataset, ranging from 275 to 1,517 cells per individual zebrafish (Table S2 and Fig S7A). At the cluster level, there was wide variation in assigned doublets ranging from 0.8 to 30.2% of cells within a cluster (Fig S7B). When pooling samples, a common assumption is that each cluster is composed of cells from all biological replicates. This assumption held true for nine of the identified 21 clusters in this sample, predominantly those with the most cells. The other 12 clusters were composed of cells from less than 30 replicates (animals), breaking the common assumption and suggesting a possible source of artifacts in interpreting results from these clusters (Fig S7B). This analysis could not be validated because this dataset did not contain a distinct cell-labeling method to enable orthogonal non-SNP–based demultiplexing and thereby a ground truth for comparison. Two pieces of evidence point towards the reliability of these results (1) the high accuracy of assignments from all the in silico pooled datasets and (2) that souporcell fails to provide assignments in the inbred mouse datasets (see previous subheading). Altogether,

this suggests that souporcell may be a viable means to identify biological replicates in large pools, which will provide informative metadata and enhance analysis for future experiments.

### Successful SNP–based demuxing of pooled fluorescent *Pleurodeles* samples without a genome or previous SNP information

After observing that SNP-based demultiplexing was reliable on various commonly used model species with genomes and with or without common SNP VCF files, we wanted to test the limits of what resources are required for accurate SNP-based demultiplexing. To do this, we took the current minimum adequate reference to map single-cell sequencing data, a de novo transcriptome, and did not use a common SNP VCF files. The Spanish ribbed newt, *Pleurodeles waltl*, is ar reemerging regenerative model organism, which has a high-quality de novo transcriptome (Matsunami et al, 2019) and more recently a high-quality genome (Brown et al, 2022, *Preprint*) but no available common SNPs VCF file. We first set out to assess souporcell demux assignments on pooled splenocytes from three transgenic *Pleurodeles* newts, which express different fluorescent proteins under the same ubiquitous promoter (CAG) (Joven et al, 2018; Eroglu et al, 2022). We designed this experiment to only contain non-erythroid spleen cells from one individual of each transgenic newt line, making it technically feasible to benchmark souporcell cell assignments for individuals through comparisons to fluorescent-based assignments (Fig 3A).

As performed in the *Xenopus* analysis, we selected only cells that had sufficient read depth and fluorescent gene detection for benchmarking (Figs 3B and S8A–H). We found that fluorescent-based and souporcell assignments show a high degree of correlation (Fig 3C and D). The fluorescent-based approach assigns many cells as "negative" (63% of cells pre-filter, 29% of cells after filtering) (Figs 3C and S8H). In this dataset, we observed that cells sequenced to a higher depth showed higher agreement between souporcell cell assignments and the fluorescent-based ground truth, which is expected because both methods should improve with more information (Fig S8I). Furthermore, though these samples were pooled and sequenced as one sample, we find dramatic variation between individuals in the *Pleurodeles* splenocyte data (i.e., clusters derived from primarily one animal). The heterogeneity of sample representation in different cell clusters highlights the critical need for demultiplexing of pooled scRNA-seq data (Fig S8A–C, J, and K). Without demultiplexing, erroneous conclusions on novel cell states or types may arise.

We found a high degree of agreement between fluorescent-based cell assignments and SNP-based assignments from *Pleurodeles* scRNA-seq data (Fig 3E). Of cells assigned by fluorescent-based demuxing to one of the three transgenic animals, 78–93% of those cells were correctly identified by souporcell (Fig 3F). When the two methods disagree, the most prevalent occurrence is "negative" fluorescent–based cell assignments that souporcell assigned to one of the transgenic animals (i.e., the rescue we also found in the *Xenopus* data). Similar to the *Xenopus* dataset, we attribute these fluorescence-based "negative" assignments to the low capture of fluorescent gene reads (Figs 3B and C and S8A and C). The next most common discrepancy between the two demultiplexing approaches

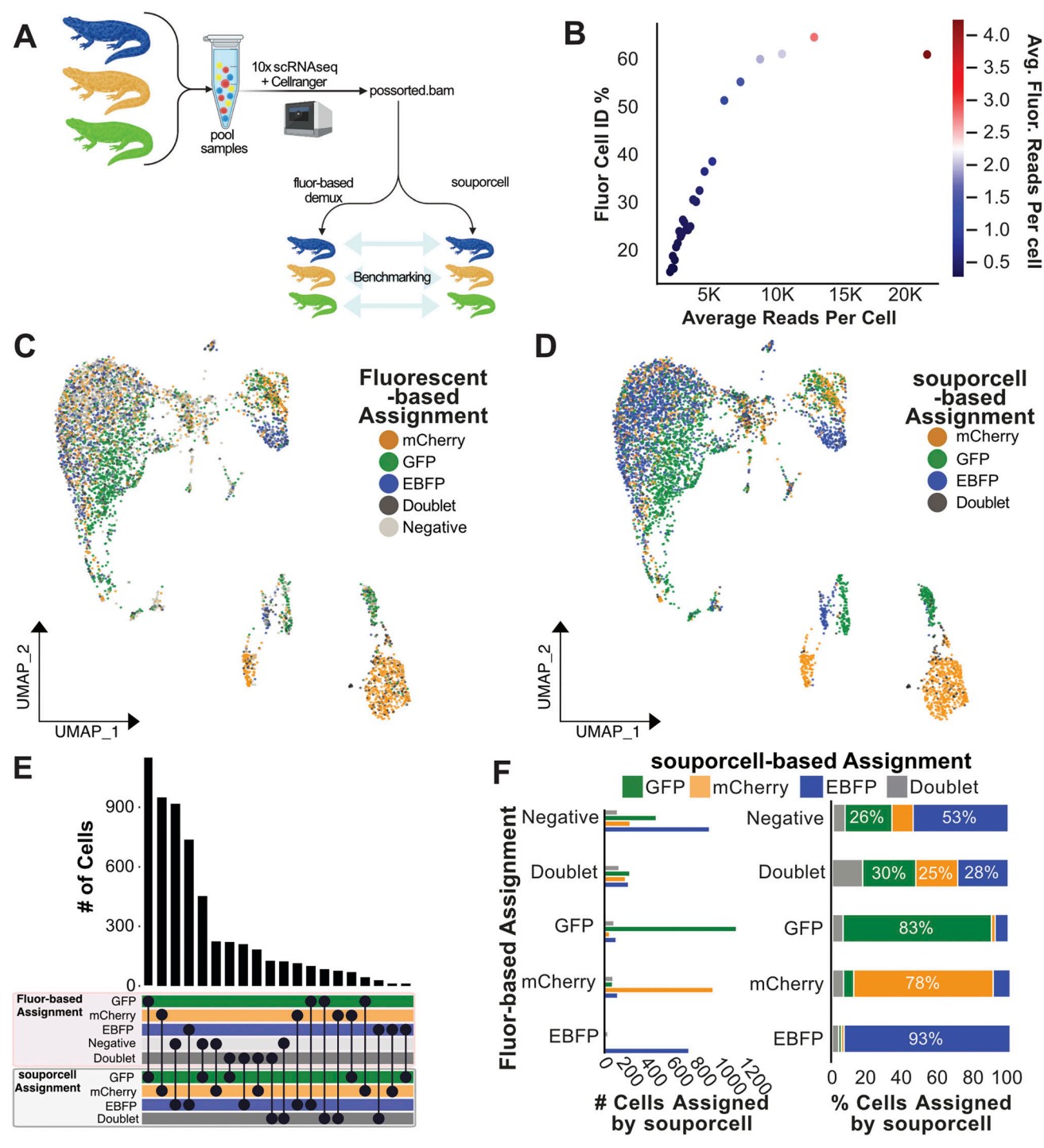

**Figure 3.  Experimentally pooled *Pleurodeles* scRNA-seq from fluorescently expressing transgenic animals accurately demultiplexed by souporcell.**
**(A)** Conceptual diagram of benchmarking analysis for this Pleurodeles dataset (n = 3). **(B)** Cell identification percentage (Fluor Cell ID %) by fluorescent-based assignment is plotted against average read depth. All cells were sorted by read depth, and binned into 40 groups before calculating Fluor Cell ID %, average total, and fluorescent read depth. Fluor Cell ID % is defined as the percentage of cells in each bin that were assigned to any of the three transgenic animals by fluorescent-based demultiplexing analysis. Binned data are also colored by the average number of summed fluorescent reads per cell. Subsequent analysis plots focused on cells with between 5,000 and 40,000 mapped reads, and >0 summed fluorescent gene reads. **(C)** UMAP plot of *Pleurodeles* pooled scRNA-seq data colored by fluorescent-based cell assignments. **(D)** UMAP plot of Pleurodeles pooled scRNA-seq data colored by souporcell assignments relabelled according to correlating transgenic animal line. **(E)** Upset plot comparing cell assignments by souporcell- to fluorescent-based assignments. **(F)** Bar plots quantifying the distribution of souporcell assignments for cells from each animal. Left: of cells assigned to each transgenic animal through fluorescent-based assignments, the number of those cells assigned by souporcell to each animal is plotted. Right: of cells assigned to each transgenic animal through fluorescent-based assignments, souporcell assignments are shown as a percentage of total cells in that category.

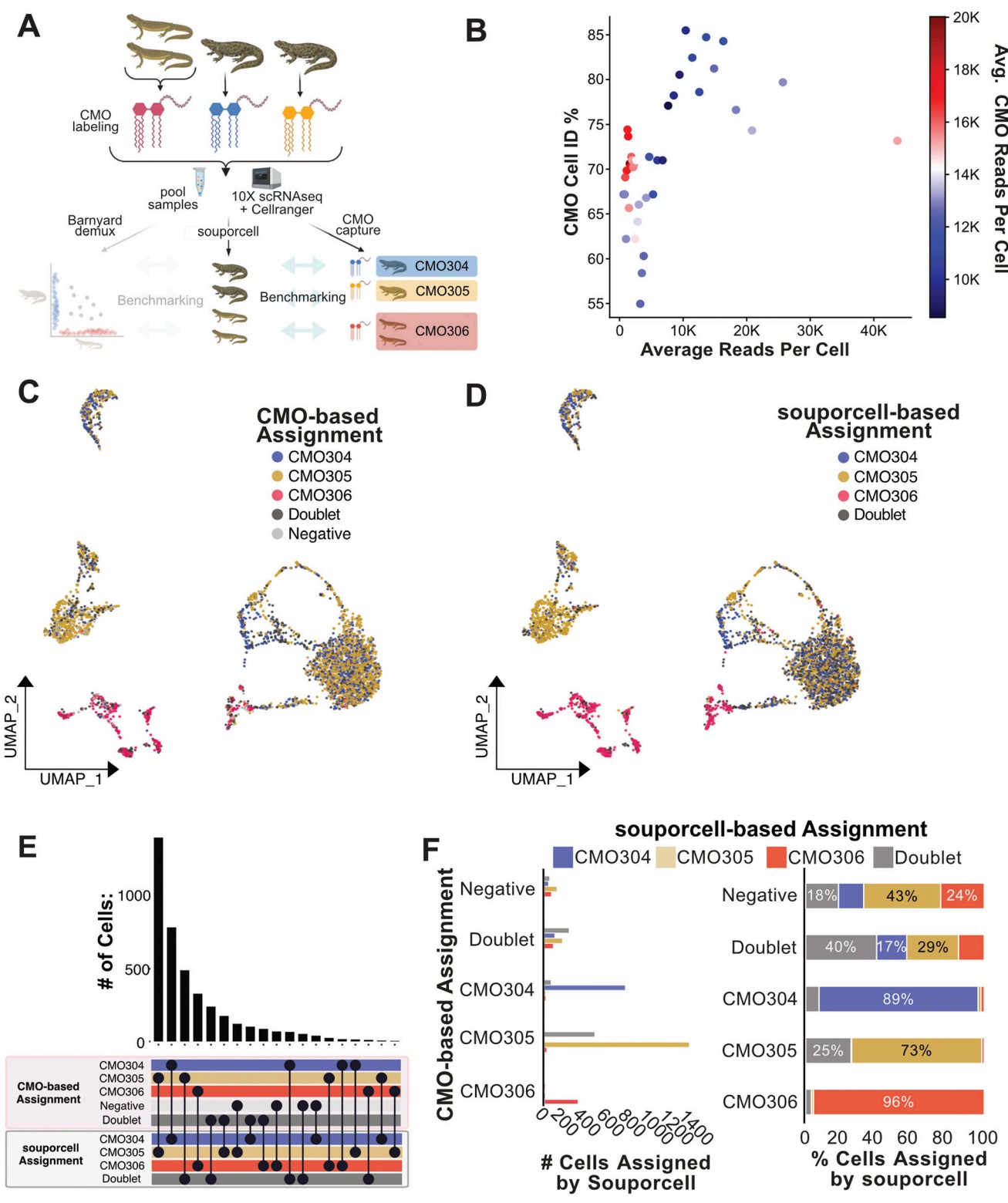

**Figure 4. Lipid-linked CMO-based demultiplexing of two salamander species, four animal-pooled scRNA-seq dataset.**
**(A)** Conceptual diagram of benchmarking analysis for this multispecies dataset (n = 4). **(B)** Cell identification percentage (CMO Cell ID %) by CMO-based assignments is plotted against average read depth. All cells were sorted by read depth, and binned into 40 groups before calculating CMO Cell ID %, and average total read depth per cell. CMO Cell ID % is defined as the percentage of cells in each bin that were any of the three CMO groups via CMO analysis. Subsequent analysis plots focused on high accuracy cells with between 5,000 and 40,000 mapped reads. **(C)** UMAP plot of Pleurodeles and Notophthalmus pooled scRNA-seq data colored by CMO assignments. **(D)** UMAP plot of Pleurodeles and Notophthalmus pooled scRNA-seq data colored by souporcell assignments relabelled according to correlating CMO labels. **(E)** Upset plot

were contested doublets, cells labeled as "doublet" by fluorescent-based demuxing, for which souporcell disagrees. The low number of mapped fluorescent reads in these samples means that a "doublet" assignment by fluorescent-based demuxing likely indicates that a particular cell had one to two counts of two distinct fluorescent genes. This could be indicative of a doublet, but it is also possible that these are erroneous doublet assignments because the fluorescent-based doublet assignments too often rely on sparse data.

### SNP-based demultiplexing succeeds in two species, pooled single-cell RNA-seq as shown by benchmarking against lipid hashing

We next wanted to determine whether SNP-based demultiplexers could succeed in demuxing scRNA-seq datasets containing cells pooled from multiple species. This multi-species approach would be particularly useful for cross-species analyses and ecological studies. This approach capitalizes on the so-called "barnyard" approach, that is typically used when investigating doublet rates in new single-cell technologies. For this experiment, we pooled splenocytes from two *P. waltl* and two *Notophthalmus viridescens* (Fig S9A). *Notophthalmus* is another salamander species studied for its regenerative capacity but equipped with only a de novo transcriptome and no published common SNP VCF file. To investigate SNP-based assignments and doublets in this dataset, we applied souporcell and a barnyard style analysis to this dataset and found a general agreement between these two approaches (Fig S9A–G).

Within this experiment of dual-species pooled 10x Genomics scRNA-seq libraries, distinct lipid hashing CMO labels were applied to label the origin of each of the *Pleurodeles* samples, and one hashing label was used for the pooled *Notophthalmus* samples, totaling three hashing labels (Fig 4A). This approach provided a means to benchmark souporcell against the current best practice and the only available commercial multiplexing strategy for cells from species without specific hashing antibodies. For the analysis, sample identity determined by applying MULTIseqDemux to CMO data was used to evaluate SNP-based demultiplexing results. We constructed and aligned to a dual-species index made from the SuperTranscriptome (Abdullayev et al, 2013; Davidson et al, 2017; Matsunami et al, 2019) for each of these species. We removed low quality, low-depth cells for further benchmarking analysis and identified which souporcell individual assignments corresponded to each CMO through analyzing the correlation of assignments between the two methods (Fig 4B). Interestingly, we did see that cells with more biological read depth showed improved percent agreement between souporcell- and Cellplex-based assignments, which was not a given because the Cellplex label is a distinct library (Fig S10). This suggests that SNP-based demuxers do display a performance relationship with increased read depth. One important

note, it appeared that many cells with high CMO reads per cell have relatively low biological reads per cell (Fig 4B), suggesting that although these cells can be CMO demultiplexed they would be unlikely to provide adequate biological information.

We found that souporcell sample assignments correlated tightly with CMO-based assignments (Fig 4C–E). When CMO-based assignments were interpreted as ground truth, the accuracy of souporcell assignments was clear, with 73–96% agreement between the two methods for assignment to the three CMO-based assignments (Fig 4F). The largest difference in assignments between these methods was that souporcell identified many doublets in the data from one CMO-labeled sample. SNP-based demultiplexers can only detect doublets from two different individuals (i.e., heterotopic doublets) which occurs at increasing ratios as sample number in the pool increases (e.g., three samples = 66% heterotypic doublets, four samples 75% heterotypic doublets, etc.). We thus also tested a second means to detect doublets, transcriptional-based doublet detection using scds to determine if this could compensate for missed doublets assignments (Bais & Kostka, 2020). We found that scds assigned many doublets not assigned by souporcell and vice versa (Fig S11). Overall, souporcell and scds agreed on 68% of assignments, including singlets and doublets. In relation to Cellplex-based assignments, both computational approaches are in high agreement, but assign doublet identities to some Cellplex-based singlet assignments. This suggests that transcriptional-based doublet detection potentially identifies doublets missed by souporcell and wet laboratory–based approaches. In general, these analyses demonstrate high agreement between CMO-based and SNP-based demultiplexing of this data from a notably complex experiment using species with poor genomic resources. Altogether, these data suggest that SNP-based demultiplexing is a powerful and near universal method to demultiplex pooled samples.

## Discussion

The commercial appearance of single-cell sequencing technologies has enabled the study of complex tissues from any species. The technical and financial hurdles posed by these technologies can discourage their use, especially when biological replicates are needed to produce reliable datasets. In the coming years, we expect that similar to bulk RNA sequencing (Schurch et al, 2016), increased biological replicates should be prioritized over increased information on one sample in sc-seq studies. We demonstrate that sample pooling is a useful and viable tool for empowering single-cell datasets. It is critical that pooling and demultiplexing approaches are also species agnostic, as one of the major advantages of scRNA-seq is its broad applicability in most organisms.

The results presented in this study are in strong support of using sample pooling and SNP-based demultiplexers in scRNA-seq studies in species which possesses between individual genetic variability. We show with numerous species which have varying

comparing cell assignments by souporcell to CMO-based assignments. **(F)** Bar plots quantifying the distribution of souporcell assignments for cells from each CMO group. Left: of cells assigned to each CMO group, the number of those cells assigned by souporcell to each identity is plotted. Right: of cells assigned to each CMO group, souporcell assignments are shown as a percentage of total cells in that category.

genomic resources and genetic variability that SNP-based demultiplexing produces highly accurate demuxing results. We used a diverse swath of methods, each with unique strengths, to validate the widespread use of SNP-based demultiplexing. We attempted to estimate the SNP density required for successfully demultiplexing, but recommend in other species a pilot experiment in which a secondary demultiplexing method, or in silico pooling, is used to validate SNP-based demux results. This will also allow for the fine tuning of parameters to ensure trustworthy results. For example, testing out a low-quality common SNPs VCF from a more obscure model organism versus a sample-derived VCF may demonstrate that the sample-derived VCF performs better.

It will also be important to define the upper limit for the number of distinct pooled samples followed by SNP-based demuxing for each organism. In line with this, we obtained assignments when performing SNP-based demultiplexing on a pool of 30 zebrafish samples. However, without a benchmarking assessment from a ground truth derived from a distinct technology, it is uncertain if these results can be trusted. We therefore propose that using SNP-based demultiplexers on large pools needs to be further validated. This can be performed in silico as more single-replicate scRNA-seq datasets become published. Until then, the developers of souporcell indicate that 21 pooled human samples can be demultiplexed and speculate that this could work in up to 40 (https://github.com/wheaton5/souporcell/issues/30). It is more likely that these large pool experiments occur in non-human samples. Thus, to aid future validations, we provide examples and modified memory-efficient scripts to pool samples and determine accuracy which will aid laboratories' working in any species to conduct their own benchmarking. In addition, this memory-efficient script allows for pooling without a VCF file, which will be critical for all researchers interested in benchmarking SNP demuxers in organisms without VCF files available. Once upper limits become established, another option to increase throughput would be layered multiplexing, for example, labeling 10 individuals with a CMO, another 10 with a second CMO, and so on. This could be paired with a second SNP-based demultiplexing step and could substantially expand sample throughput.

The ability to dramatically increase biological replicate throughput will have important implications for the quality of scRNA-seq data. Animal-to-animal variability in gene expression patterns has been observed for decades and is expected in many tissues, especially in immune cells, of animals that encounter different stressors or pathogens (Boedigheimer et al, 2008; Schokker et al, 2015; Paunovska et al, 2018; Reid et al, 2018). The existence of this natural biological variability coupled with technical variability in single-cell methods poses problems for studies without replicates or those that do not demultiplex pooled datasets (Hicks et al, 2018). Demultiplexing of pooled scRNA-seq is required to associate metadata with individual samples to account for between sample variation, replicate strengthened differential expression analysis, and confirmation of sample representation across scRNA-seq datasets and cell clusters (Squair et al, 2021; Zimmerman et al, 2021). The potential for heterogeneity in sample representation across cell clusters can be clearly seen in Figs 3 and S8, where some cell clusters are from one or two of the three individuals present in the pool. Although a variety of

batch correction algorithms are available for single-cell studies (Tran et al, 2020), these cannot be used if there are no metadata distinguishing replicates. We recommend that SNP-based demultiplexing become a standard quality control for all pooled samples from animals with between animal genetic diversity.

An additional benefit of pooled single-cell experiments is the superloading of cells followed by heterotypic doublet detection. However, our results from synthetic pooled samples saw varied success in heterotypic doublet detection by SNP-based demultiplexers. Our analyses also indicate that sample number in each pool may directly impact doublet-detection efficiency, which should be considered in experimental design. We argue that the most pragmatic approach for doublet detection includes a combination of SNP-based and transcriptomic-based methods. In line with this, a thorough benchmarking in human samples of transcriptional and SNP-based doublet detectors suggest that an intersectional approach to doublet detection is superior to any one single method (Neavin et al, 2022 Preprint). Multiple independent programs designed specifically for doublet detection using transcriptomic instead of genotypic information are available for scRNA-seq data including DoubletDetection (Gayoso & Shor, 2022), Solo (Bernstein et al, 2020), and Scds (Bais & Kostka, 2020). To gain the full benefits of SNP-based demultiplexing with robust transcriptomic doublet detection, we recommend using one or more of these independent doublet-detection programs to identify and remove cell barcodes likely to contain doublets. We anticipate that optimized doublet detection algorithms along with improved bioinformatic resources for each species would improve doublet detection and facilitate superloading pooled single-cell data.

Many bioinformatic tools for scRNA-seq are not easily applied to all species. We intentionally applied a SNP-based demuxer to experimental set ups that would traditionally be expected to be computationally challenging, demonstrated the successful application, and provide a table and flowchart to illustrate the workarounds used in each case (Figs S2 and S12). Fortunately, we demonstrate that SNP demuxing can be performed without resources like a high quality, well annotated genome, or a population-wide common SNP genotypes VCF file. This is where souporcell stood out as the only tool that was feasible to use with only a de novo transcriptome and no VCF input. A secondary hurdle for applying SNP-based demuxers is that these tools often struggle when applied to datasets from species with large reference genomes or de novo transcriptomes (which typically have many contigs). When using de novo transcriptomes (i.e., Pleurodeles or Notophthalmus), we modified the default souporcell pipeline to enable remapping of reads (Fig S12, see the Materials and Methods section). Finally, though pooling of multiple species into one experiment is not currently commonplace, we demonstrated the successful SNP-based demuxing of a pooled two-species experiment. We expect that variations in cell or nuclei sizes between species could cause biases in cell capture depending on the scRNA-seq library preparation method, especially with microfluidic scRNA-seq. Despite these potential challenges, this experiment demonstrates the success of SNP-based demuxing in a particularly challenging scenario, and as more research focuses on cross-species comparisons, this is a clever means to increase sample throughput.

Overall, we successfully applied SNP-based methods to demultiplex pooled single-cell data from multiple species and a two species mix. Our benchmarking results suggest that SNP-based demultiplexing in these species is accurate relative to other available demultiplexing approaches. We hope that this study will increase awareness of single-cell pooling and SNP-based demultiplexing approaches for research communities not yet using these methods. Including SNP-based demuxers in experimental designs for future (and past) studies will greatly expand single-cell–based discoveries. This will facilitate work in well-known and lesser studied species by lowering the financial and technical hurdles of producing adequately powered single-cell experiments. We predict that both species agnostic and cross-species comparative studies are going to be increasingly fruitful in uncovering biological insights and the application of SNP-based demultiplexing with minimal genomic resources is critical for future research.

# Materials and Methods

### Animal handling and ethics

All experiments were carried out in post-metamorphic *P. waltl* and *N. viridescens* at Karolinska Institutet and were performed according to local and European ethical permits. *N. viridescens* and *P. waltl* were raised in-house. All animals were maintained under standard conditions of 12-h light/12-h darkness at 18–24°C (Joven et al, 2015). Before all experiments, animals were fully narcotized in 0.1% tricaine in housing water. For *Pleurodeles* in the fluorescent pool, experimental animals were housed in carbon-supplemented filtered tap water (55 g Tetra Marine Sea Salt, 11 g Ektozon N Salt, 2.5 ml of water conditioner/dechlorinator (Seachem Prime - Vattenberedningsmedel), 20 ml of Yokuchi Bitamin Multivitamin and 10 ml of calcium supplementation (Easy-Life Calcium) into 100 Liters of water). For *Pleurodeles* and *Notophthalmus* in the Cellplex experiment, animals were housed in the water as described but modified to only have sea salt, Ektozon, and calcium solution.

### Collection of splenocytes for scRNA-seq

#### Fluorescence-pooling experiment
Spleens were harvested from three separate *P. waltl* and processed as individual samples in parallel. All animals were post-metamorphic newts from established transgenic lines close to sexual maturity: one female tgTol2(CAG:Nucbow CAG:Cytbow)[Simon] (5.67 g weight, 10.8 cm snout-to-tail length) (Joven et al, 2018), one male tgSceI(CAG:loxP-GFP-loxP-Cherry)[Simon] (5.36 g, 11.1 cm) (Joven et al, 2018), and female tgTol2(CAG:loxP-Cherry-loxP-H2B:YFP)[Simon] (6.25 g, 10.8 cm) (Eroglu et al, 2022). Forceps and iridectomy scissors were used to remove the spleen in one piece, making sure that the forceps did not tear the spleen. Iridectomy scissors were used to carefully remove connective tissue. A 70-$\mu$m nylon mesh filter was inserted into a 50 ml conical, and 1 ml of ice cold 0.7X PBS was added to pre-wet the filter. The spleen was placed on the pre-wetted filter and slowly mashed through the filter using the plunger stopper end of a 3-ml syringe. Once the spleen appeared translucent, the plunger and filter were thoroughly washed with ~10 ml ice cold 0.7X PBS and making sure no PBS was left on the plunger or filter. The 0.7X PBS solution was then poured into a 15 ml conical and centrifuged at 300$g$ for 5 min at 4°C in a swinging bucket rotor. Supernatant was decanted, and cells were resuspended in 1 ml of 0.7X PBS. Cells were then counted using trypan blue to assess viability. Viabilities of non-erythroid cells (based on cellular morphology) were 85%, 98.4%, and 91.6% in eBFP, eGFP, and mCherry animals, respectively. FACS was used to sort for fluorescent-positive cells (Fig S13A–D). The samples have been analyzed without any removal of red blood cells using a BD Influx cytometer; the selected cells were sorted using a 100-$\mu$m nozzle in bulk into 1.5 ml Eppendorf tubes. The cell preparations and the sorted cells were kept in 4°C throughout the sorting. The results were analyzed with FlowJo software 10.8.1.

Debris and erythrocytes (note: erythrocytes do not express fluorescent markers under the CAG promoter and are far larger than other splenocytes) (Fig S13E) were excluded with the gating strategy in the side scatter–forward scatter and singlet discrimination plots revealing separated fluorescent subpopulations of GFP, mCherry, and BFP as confirmed by sorting on microscopy slides (Figs S13 and S14). We sorted the subpopulations with the highest expression levels of each fluorescent tag. In total, $4 \times 10^5$ cells of GFP+, $4 \times 10^5$ of mCherry+, and $3.15 \times 10^5$ BFP+ were isolated in individual 1.5 ml Eppendorf tubes. GFP and mCherry expression were high, but BFP expression was dim. 500 $\mu$l of each solution was then added to an individual 1.5 ml Eppendorf tube for the fluorescent pool sample (trial mixture of pooled cells shown Fig S13D). This was centrifuged in a 1.5 ml Eppendorf at 4°C at 300$g$ on a tabletop centrifuge. The supernatant was carefully removed and resuspended in the remaining volume. Cells were then manually counted and adjusted to a concentration targeting the collection of 10,000 cells on the 10x Genomics controller.

#### Cellplex/barnyard-pooling experiment
Spleens from one female *N. viridescens* (4.45 g and 10.6 cm snout-to-tail) and one male *N. viridescens* (3.55 g and 10.2 cm) were collected, pooled into one tube, and then processed as described in "*fluorescence-pooling experiment*" excluding FACS cell sorting. The only modification to the above described processing is that cells were kept at room temperature throughout and that cells were resuspended in 0.7X PBS with 0.04% ultrapure BSA.

For *P. waltl*, spleens were removed from animals as described in "*fluorescent-pooling experiment*" from one adult tgSceI(CAG:loxP-GFP-loxP-Cherry)[Simon] female (23.5 g and 16.1 cm snout-to-tail length), one male tgSceI(CAG:loxP-GFP-loxP-Cherry)[Simon] (13.95 g and 15.7 cm) animal. *Pleurodeles* were processed as individual samples. After the spleen was thoroughly mashed through the pre-wetted 70-$\mu$m nylon filter and the filter being washed with 10 ml of 0.7X PBS, the cells were centrifuged at 300$g$ for 5 min. Splenocytes were resuspended cells in 1 ml of sterile filtered 1X ACK (http://cshprotocols.cshlp.org/content/2014/11/pdb.rec083295.short) to lyse red blood cells. After one minute of lysis, cells were diluted with 10 ml of 0.7X PBS and filtered through a 70-$\mu$m nylon mesh filter and centrifuged at 300$g$ for 5 min. Cells were then resuspended in 0.7X PBS with 0.04% ultrapure BSA.

The pooled *Notophthalmus* sample and the individual *Pleurodeles* samples were then taken through the 10x Genomics 3′ Cellplex–labeling protocol (CG000391; Demonstrated Protocol) the only modifications being the use of 0.7X PBS + 0.04% BSA for all wash and resuspension steps. Samples were stained with CM304 (*Pleurodeles* female), CMO305 (*Pleurodeles* male), and CMO306 (pool of *Notophthalmus* samples, one male and one female). Samples were manually counted and pooled at equal ratios immediately before loading onto the 10x Genomics Chromium Controller targeting 9,000 cells in total.

## Preparation and sequencing of single-cell RNA sequencing libraries

Chromium single-cell 3′ kit v3 (10x Genomics) was used according to the manufacturer's instructions.

## Generation of SuperTranscriptome and corresponding gtf files

A *P. waltl* de novo transcriptome from the study of Matsunami et al (2019) was downloaded from https://figshare.com/articles/dataset/Trinity_Pwal_v2_fasta_gz/7106033/1 and unzipped. The Trinity (Haas et al, 2013) singularity image v2.11.0 was then used to generate a *P. waltl* SuperTranscriptome like so:

singularity exec -e trinityrnaseq.v2.11.0.simg /usr/local/bin/trinityrnaseq/Analysis/SuperTranscripts/Trinity_gene_splice_modeler.py –trinity_fasta Trinity_Pwal_v2.fasta –incl_malign –out_prefix out_dir

A *Notophthalmus* de novo transcriptome from the study of Abdullayev et al (2013) was downloaded from https://sandberglab.se/static/data/papers/redspottednewt/reference_transcripts_v2.fa.gz and decompressed. Transcripts were clustered with cd-hit-est (Li & Godzik, 2006; Fu et al, 2012).

cd-hit-est -o s -d -c 0.98 -i reference_transcripts_v2.fa -p 1 -d 0 -b 3 -T 20 -M 20000000.

The resulting cdhit.clstr file was then parsed using clstr2txt.pl from the cdhit package to generate a clusters.txt file. The id and clstr columns were obtained from this file and used to make an info.clusters.txt file to use with Lace (Davidson et al, 2017). Lace v1.14.1 was run: Lace_run.py reference_transcripts_v2.fa info.clusters.txt -t –cores 16 -o Noto_superTrans. The resulting gff files from both SuperTranscriptomes were converted to a gtf using AGAT perl script agat_convert_sp_gff2gtf.pl (Dainat et al, 2022).

## Zebrafish demultiplexing

FASTQ files from a previously published study (Rubin et al, 2022) (SRA accessions: SRR17218111, SRR17218112, SRR17218113, SRR17218114, SRR17218091, SRR17218092) were downloaded from SRA using prefetch followed by fasterq-dump with flags split-files and include-technical. Files were renamed to 10x FASTQ format (e.g., sample_S1_L001_I1_001.fastq.gz) and then aligned using Cell Ranger v7.0.0 count to GRCz11 with the corresponding gtf for GRCz11 filtered via Cell Ranger mkgtf for protein_coding genes.

To merge bams in silico, a VCF file and tbi index were downloaded from https://research.nhgri.nih.gov/manuscripts/Burgess/zebrafish/downloads/NHGRI-1/danRer11/danRer11Tracks/NHGRI1.danRer11.variant.vcf.gz and https://research.nhgri.nih.gov/manuscripts/Burgess/zebrafish/downloads/NHGRI-1/danRer11/danRer11Tracks/NHGRI1.danRer11.variant.vcf.gz.tbi (LaFave et al, 2014) and subsequently filtered using bcftools v1.16. For ease of use, we used the well-maintained and annotated Demuxafy singularity container, which contains almost all of the demuxers and other required bioinformatic tools, for as many analyses as possible (Neavin et al, 2022 *Preprint*).

singularity exec Demuxafy.sif bcftools filter –include "MAF ≥ 0.05" -Oz –output NHGRI1.maf0.05.danRer11.variant.vcf.gz NHGRI1.danRer11.variant.vcf.gz

singularity exec Demuxafy.sif bcftools sort -Oz NHGRI1.maf0.05.danRer11.variant.vcf.gz -o sorted.NHGRI1.maf0.05.danRer11.variant.vcf.gz

The chromosomes between the vcf and gtf did not match so bcftools annotate –rename-chrs was used to change chromosome names in the VCF using a tab-separated file named chr.conv.txt with the format: chr1 1, chr2 2, etc.

singularity exec Demuxafy.sif bcftools annotate –rename-chrs chr.conv.txt sorted.NHGRI1.maf0.05.danRer11.variant.vcf.gz | singularity exec Demuxafy.sif bgzip > rename.sorted.NHGRI1.maf0.05.danRer11.variant.vcf.gz

The sample-specific bam outputs were then merged using Vireo's synth_pool.py script as follows:

python synth_pool.py -s sample1_genome_bam.bam,sample2_genome_bam.bam,sample3_possorted_genome_bam.bam -b sample1/outs/filtered_feature_bc_matrix/barcodes.tsv,sample2/outs/filtered_feature_bc_matrix/barcodes.tsv,sample3/outs/filtered_feature_bc_matrix/barcodes.tsv -d 0.1 -o three_mixed_zf -p 1 -r NHGRI1.maf0.05.danRer11.variant.vcf.gz –randomSEED 50 –nCELL 500.

Souporcell was run using souporcell_pipeline.py with inputs: merged BAM output from synth_pool.py, the output barcodes_pool.tsv from synth_pool.py, the genome fasta (Danio_rerio.GRCz11.dna.primary_assembly.fa), N = 3, and vcf file NHGRI1.maf0.05.danRer11.variant.vcf.gz.

### Souporcell

singularity exec Demuxafy.sif souporcell_pipeline.py -i pooled.bam -b barcodes_pool.tsv -f Danio_rerio.GRCz11.dna.primary_assembly.fa -t 20 -o output -k 3 –common_variants NHGRI1.maf0.05.danRer11.variant.vcf.

### Freemuxlet

We found that Freemuxlet failed when using the downloaded VCF file but was successful when inputting the VCF and minimap.bam generated via souporcell when running souporcell without a VCF. This implies that Freemuxlet may be able to be fed a sample-derived VCF (and that this may be more reliable than a low-quality common SNPs VCF file). An example of a VCF that worked:

singularity exec Demuxafy.sif freebayes -f Danio_rerio.GRCz11.dna.primary_assembly.fa -iXu -C 2 -q 20 -n 3 -E 1 -m 30 -g 100,000 souporcell_minimap_tagged_sorted.bam > zf.freebayes.vcf

Then Freemuxlet:

singularity exec Demuxafy.sif popscle dsc-pileup –sam pooled.bam –group-list barcodes_pool.tsv –vcf zf.freebayes.vcf –out $FREEMUXLET_OUTDIR/pileup

singularity exec Demuxafy.sif popscle freemuxlet –plp $FREEMUXLET_OUTDIR/pileup –out $FREEMUXLET_OUTDIR/freemuxlet –group-list barcodes_pool.tsv –nsample 3

singularity exec Demuxafy.sif bash Freemuxlet_summary.sh $FREEMUXLET_OUTDIR/freemuxlet.clust1.samples.gz > $FREEMUXLET_OUTDIR/freemuxlet_summary.tsv
Vireo:
singularity exec Demuxafy.sif samtools idxstats pooled.bam > chromosomes.txtawk "{print $1}" chromosomes.txt | paste -s -d, > chr.list.vireo.txt

singularity exec Demuxafy.sif cellsnp-lite -s pooled.bam -b barcodes_pool.tsv -O $OUT_DIR -p 20 –chrom "$(<${DemuxSoupDir}chr.list.vireo.txt)" –minMAF 0.1 –minCOUNT 100 –gzip

singularity exec ${DemuxSoupDir}Demuxafy.sif vireo -c $OUT_DIR -o $OUT_DIR -N $N.
scSplit:
singularity exec Demuxafy.sif samtools view -b -S -q 10 -F 3844 pooled.bam > $SCSPLIT_OUTDIR/filtered_bam.bam
singularity exec Demuxafy.sif samtools rmdup $SCSPLIT_OUTDIR/filtered_bam.bam $SCSPLIT_OUTDIR/filtered_bam_dedup.bam
singularity exec Demuxafy.sif samtools sort -o $SCSPLIT_OUTDIR/filtered_bam_dedup_sorted.bam $SCSPLIT_OUTDIR/filtered_bam_dedup.bam
singularity exec Demuxafy.sif samtools index $SCSPLIT_OUTDIR/filtered_bam_dedup_sorted.bam

singularity exec Demuxafy.sif freebayes -f Danio_rerio.GRCz11.dna.primary_assembly.fa -iXu -C 2 -q 1 $SCSPLIT_OUTDIR/filtered_bam_dedup_sorted.bam > $SCSPLIT_OUTDIR/freebayes_var.vcf
singularity exec Demuxafy.sif vcftools –gzvcf $SCSPLIT_OUTDIR/freebayes_var.vcf –minQ 30 –recode –recode-INFO-all –out $SCSPLIT_OUTDIR/freebayes_var_qual30

singularity exec Demuxafy.sif scSplit count -c NHGRI1.maf0.05.danRer11.variant.vcf -v $SCSPLIT_OUTDIR/freebayes_var_qual30.recode.vcf -i $SCSPLIT_OUTDIR/filtered_bam_dedup_sorted.bam -b barcodes_pool.tsv -r $SCSPLIT_OUTDIR/ref_filtered.csv -a $SCSPLIT_OUTDIR/alt_filtered.csv -o $SCSPLIT_OUTDIR
singularity exec Demuxafy.sif scSplit run -r $SCSPLIT_OUTDIR/ref_filtered.csv -a $SCSPLIT_OUTDIR/alt_filtered.csv -n 3 -o $SCSPLIT_OUTDIR
singularity exec Demuxafy.sif scSplit genotype -r $SCSPLIT_OUTDIR/ref_filtered.csv -a $SCSPLIT_OUTDIR/alt_filtered.csv -p $SCSPLIT_OUTDIR/scSplit_P_s_c.csv -o $SCSPLIT_OUTDIR

singularity exec Demuxafy.sif bash scSplit_summary.sh $SCSPLIT_OUTDIR/scSplit_result.csv.
A previously published dataset of 30 pooled zebrafish embryos (Metikala et al, 2021) was downloaded from SRA (SRR12067711–SRR12067712) using prefetch followed by fasterq-dump with flags split-files and include-technical. Files were renamed to 10x FASTQ format (e.g., sample_S1_L001_I1_001.fastq.gz) and then aligned using Cell Ranger v7.0.0 count to GRCz11 with the corresponding gtf for GRCz11 filtered via Cell Ranger mkgtf for protein_coding genes. The output possorted_genome_bam.bam and filtered barcodes.tsv file were then used to run souporcell:
singularity exec Demuxafy.sif souporcell_pipeline.py -i possorted_genome_bam.bam -b barcodes.tsv -f Danio_rerio.GRCz11.dna.primary_assembly.fa -t $THREADS -o $SOUPORCELL_OUTDIR -k 30 –common_variants NHGRI1.maf0.05.danRer11.variant.vcf
The clusters.tsv file output from souporcell was then used to evaluate cluster distribution of cells. Code for this is the basis for Figs S6 and S12 is here: https://github.com/RegenImm-Lab/SNPdemuxPaper.

### *Xenopus* souporcell demultiplexing

The Cell Ranger BAM file (https://sra-pub-src-2.s3.amazonaws.com/SRR13600554/107606_Xen_Pool_BL7_10_14dpa.bam.1) from a previously published publicly available dataset (Lin et al, 2021) of fluorescent-expressing *X. laevis* cells pooled from eight animals: two blastemas of two siblings (CAGGs:Venus), three blastemas of three siblings (CAGGs:mCherry, B51), and three samples from three siblings (CAGGs:TFPnls, G48) were download directly. SAMtools (Li et al, 2009) was used to index the BAM before running default souporcell pipeline with the barcodes file (https://ftp.ncbi.nlm.nih.gov/geo/samples/GSM5057nnn/GSM5057661/suppl/GSM5057661_107606_Xen_Pool_BL7_10_14dpa_barcodes.tsv.gz), *X. laevis* genome FASTA (https://sra-pub-src-2.s3.amazonaws.com/SRR13600553/Xenbase_v9.2.fa.1), and N = 8. Note: this specific reference includes the plasmid sequences necessary for mapping to fluorescent sequences.

### Axolotl in silico mixing and souporcell demultiplexing

FASTQ files from a previously published (Lust et al, 2022) axolotl single-nucleus RNA sequencing dataset were downloaded. Libraries from ArrayExpress (E-MTAB-11662) labeled "reseq" and from samples D_1, L_1, and M_1, three individual animals all run on individual wells on a 10x Genomics chip were downloaded. To make a Cell Ranger reference, the axolotl genome (AmexG_v6.0-DD) was downloaded from https://www.axolotl-omics.org/dl/AmexG_v6.0-DD.fa.gz along with a gtf (AmexT_v47-AmexG_v6.0-DD.gtf) https://www.axolotl-omics.org/dl/AmexT_v47-AmexG_v6.0-DD.gtf.gz, which required the removal of white space (i.e, sed "s/\ \[/_/g") for use with Cell Ranger (v7.0.0) mkref. Cell Ranger count was run on each library individually, resulting in three position–sorted BAM files from samples D_1, L_1, and M_1. BAM files were merged using synth_pool.py from Vireo (Huang et al, 2019) using the VCF downloaded from (http://ambystoma.uky.edu/hubExamples/hubAssembly/hub_AmexG_v6/AmexG_v6.hub_data/SNP_vcf_tracks/ddMale_to_AmexGv6.vcf.gz) which was filtered using BCFtools v1.11.
bcftools filter –include "MAF ≥ 0.05" -Oz –output ddMale.common_maf0.05.vcf.gz ddMale_to_AmexGv6.vcf and then sorted:
bcftools sort -Oz ddMale.common_maf0.05.vcf -o sorted.ddMale.common_maf0.05.vcf.gz

Barcodes.tsv files were obtained from filtered outputs of Cell Ranger count for each library. Doublet rate (-d) was set to 0.1 and –randomSEED 50.

python synth_pool.py -s /D_1/outs/possorted_genome_bam.bam,/L_1/outs/possorted_genome_bam.bam,/M_1/outs/possorted_genome_bam.bam /D_1/filtered_feature_bc_matrix/barcodes.tsv,/L_1/outs/filtered_feature_bc_matrix/barcodes.tsv,/M_1/outs/filtered_feature_bc_matrix/barcodes.tsv -d 0.1 -o pooled_bam -p 1 -r sorted.ddMale.common_maf0.05.vcf.gz –randomSEED 50.

Note: we only expected the troublet portion of souporcell to be capable of detecting heterotypic doublets, so for downstream analysis of this synthetically pooled data, we removed all homotypic doublets.

The pooled BAM was indexed using SAMtools index-c which made a .csi index and was renamed to have a .bai file extension for use in souporcell. Souporcell was run using souporcell_pipeline.py with inputs: merged BAM output from synth_pool.py, the output barcodes_pool.tsv from synth_pool.py, the genome fasta (AmexG_v6.0-DD.fa), N = 3, VCF file ddMale.common_maf0.05.vcf.gz, and –skip_remap SKIP_REMAP.

singularity exec Demuxafy.sif souporcell_pipeline.py -i pooled.bam -b barcodes_pool.tsv -f AmexG_v6.0-DD.fa -t 20 -o output -k 3 –skip_remap SKIP_REMAP –common_variants ddMale.common_maf0.05.vcf

### Green monkey (*Chlorocebus aethiops*) in silico mixing and demultiplexing

Five samples were downloaded from a previously published dataset (Speranza et al, 2021). Data were downloaded from SRA using prefetch followed by fasterq-dump with flags split-files and include-technical. FASTQ files were obtained from AGM1_Mediastinal Lymph Node (SRR12507774–SRR12507781), AGM3_Mediastinal Lymph Node (SRR12507790–SRR12507797), AGM5_Mediastinal Lymph Node (SRR12507806–SRR12507813), AGM7_Mediastinal Lymph Node (SRR12507822–SRR12507829), and AGM9_Mediastinal Lymph Node (SRR12507846–SRR12507853). *C. aethiops* has a robust VCF file available (European Variation Archive: PRJEB7923) that needs to be used in conjunction with genome assembly Chlorocebus_sabeus 1.1 (GCA_000409795.2). This assembly did not have an annotation file available, and we generated a gtf file for this GenBank assembly using minimap2 (Li, 2018) and the below described steps.

minimap2 -ax splice:hq -t 10 GCF_000409795.2_Chlorocebus_sabeus_1.1_cds_from_genomic.fna GCA_000409795.2_Chlorocebus_sabeus_1.1_genomic.fna | samtools sort -O BAM -o minimap2.bam.

Followed by BEDtools (Quinlan & Hall, 2010):

bedtools bamtobed -bed12 -i trans2gene.minimap2.bam > alignments.bedbedToGenePred alignments.bed alignments.genepred genePredToGtf "file" alignments.genepred GCA_000409795.2.minimap.gtf

This gtf was then used with GCA_000409795.2_Chlorocebus_sabeus_1.1_genomic.fna to generate a Cell Ranger reference using Cell Ranger mkref. Cell Ranger count was used with default settings to align the downloaded libraries. To merge the BAM outputs, we first used BCFtools (Danecek et al, 2021) to filter, sort, and change chromosomes name in the VCF:

bcftools filter –include "MAF ≥ 0.05" -Oz –output Vervet-maf0.05.500KFinal_EVA.vcf.gz Vervet500KFinal_EVA.vcf.gzbcftools sort -Oz Vervet.maf0.05.500KFinal_EVA.vcf.gz -o sorted.Vervet.maf0.05.500KFinal_EVA.vcf.gz

And then converted chromosome names in this VCF to match the chromosomes in the bam files after mapping. chr_name_conv.txt has the format of "1 CM001941.2," with 1 being the original chromosome number and CM001941.2 being the chromosome accession number; this was carried out from chromosome 1–29.

bcftools annotate –rename-chrs chr_name_conv.txt sorted.Vervet.maf0.05.500KFinal_EVA.vcf.gz | bgzip > rename.sorted.Vervet.maf0.05.500KFinal_EVA.vcf.gz

Because of low reads/cell in these libraries, we selected barcodes using Seurat with between 1,000 and 2,000 features. These filtered barcode files were used with 10x subset-bam (https://github.com/10XGenomics/subset-bam) (e.g., subset-bam –bam possorted_genome_bam.bam –cell-barcodes filtered.barcodes.tsv –out-bam filtered.bam) to create BAMs with these high quality cells. BAMs were subsequently merged using Vireo:

synth_pool.py filt.LN1.bam,filt.LN3.bam,filt.LN5.bam,filt.LN7.bam,filt.LN9.bam -b LN1.barcodes.tsv,LN3.barcodes.tsv,LN5.barcodes.tsv,LN7.barcodes.tsv,LN9.barcodes.tsv -r rename.sorted.Vervet.maf0.05.500KFinal_EVA.vcf -d 0.1 -o mixed_monkey -p 1.

Finally, cells were SNP demultiplexed using souporcell:

singularity exec Demuxafy.sif souporcell_pipeline.py -i pooled.bam -b barcodes_pool.tsv -f GCA_000409795.2_Chlorocebus_sabeus_1.1_genomic.fna -t 20 -o output -k 5 –skip_remap SKIP_REMAP –common_variants rename.sorted.Vervet.maf0.05.500KFinal_EVA.vcf

Freemuxlet:

Important note: as with the zebrafish data, we found that Freemuxlet failed when using the downloaded VCF file but was successful when inputting the VCF and minimap.bam generated via souporcell when running souporcell without a VCF. See above Freemuxlet zebrafish code for an example.

singularity exec Demuxafy.sif popscle dsc-pileup –sam pooled.bam –group-list barcodes_pool.tsv –vcf souporcell_merged_sorted_vcf.vcf.gz–out $FREEMUXLET_OUTDIR/pileup

singularity exec Demuxafy.sif popscle freemuxlet –plp $FREEMUXLET_OUTDIR/pileup –out $FREEMUXLET_OUTDIR/freemuxlet –group-list barcodes_pool.tsv –nsample5

singularity exec Demuxafy.sif bash Freemuxlet_summary.sh $FREEMUXLET_OUTDIR/freemuxlet.clust1.samples.gz > $FREEMUXLET_OUTDIR/freemuxlet_summary.tsv

Vireo:

A list of chromosomes to pass to cellsnp-lite was made:

singularity exec Demuxafy.sif samtools idxstats pooled.sorted.bam > chromosomes.txt

awk "{print $1}" chromosomes.txt | paste -s -d, > chr.list.vireo.txt.

And then cellsnp-lite and Vireo were run:

singularity exec Demuxafy.sif cellsnp-lite -s pooled.bam -b barcodes_pool.tsv –chrom "$(<${chr_list}chr.list.vireo.txt)" -O $VIREO_OUTDIR -p 20 –minMAF 0.1 –minCOUNT 100 –gzip

singularity exec Demuxafy.sif vireo -c $VIREO_OUTDIR -o $VIREO_OUTDIR -N 5.

scSplit:

```
singularity exec Demuxafy.sif samtools view -b -S -q 10 -F 3844 pooled.bam > $SCSPLIT_OUTDIR/filtered_bam.bam
singularity exec Demuxafy.sif samtools rmdup $SCSPLIT_OUTDIR/filtered_bam.bam $SCSPLIT_OUTDIR/filtered_bam_dedup.bam
singularity exec Demuxafy.sif samtools sort -o $SCSPLIT_OUTDIR/filtered_bam_dedup_sorted.bam $SCSPLIT_OUTDIR/filtered_bam_dedup.bam
singularity exec Demuxafy.sif samtools index $SCSPLIT_OUTDIR/filtered_bam_dedup_sorted.bam

singularity exec Demuxafy.sif freebayes -f GCF_000409795.2_Chlorocebus_sabeus_1.1_cds_from_genomic.fna -iXu -C 2 -q 1 $SCSPLIT_OUTDIR/filtered_bam_dedup_sorted.bam > $SCSPLIT_OUTDIR/freebayes_var.vcf
singularity exec Demuxafy.sif vcftools –gzvcf $SCSPLIT_OUTDIR/freebayes_var.vcf –minQ 30 –recode –recode-INFO-all –out $SCSPLIT_OUTDIR/freebayes_var_qual30

singularity exec Demuxafy.sif scSplit count -c $SCSPLIT_OUTDIR/freebayes_var.vcf -v $SCSPLIT_OUTDIR/freebayes_var_qual30.recode.vcf -i $SCSPLIT_OUTDIR/filtered_bam_dedup_sorted.bam  -b  barcodes_pool.tsv -r $SCSPLIT_OUTDIR/ref_filtered.csv  -a $SCSPLIT_OUTDIR/alt_filtered.csv -o $SCSPLIT_OUTDIR
singularity exec Demuxafy.sif scSplit run -r $SCSPLIT_OUTDIR/ref_filtered.csv  -a $SCSPLIT_OUTDIR/alt_filtered.csv -n $N -o $SCSPLIT_OUTDIR
singularity exec Demuxafy.sif scSplit genotype -r $SCSPLIT_OUTDIR/ref_filtered.csv  -a $SCSPLIT_OUTDIR/alt_filtered.csv -p $SCSPLIT_OUTDIR/scSplit_P_s_c.csv -o $SCSPLIT_OUTDIR

singularity exec Demuxafy.sif bash scSplit_summary.sh $SCSPLIT_OUTDIR/scSplit_result.csv
```

### *Pleurodeles* mapping and SNP-based demultiplexing

Cell Ranger 7.0.0 mkref command was used with the above listed *Pleurodeles* SuperTranscriptome FASTA and gtf files to produce a Cell Ranger compatible reference. Cell Ranger 7.0.0 count command was then used to map and count reads over the transcriptome for the three transgenic animal scRNA-seq dataset.

Souporcell (Heaton et al, 2020)-related processes were all run from a Demuxafy (Neavin et al, 2022 *Preprint*) singularity image (image version 1.0.3). The remapping and variant calling stages of souporcell were run externally because of problems with timeouts on the remapping process with the large salamander transcriptome, and issues with the souporcell internal freebayes command failing. The VCF from freebayes was then used in souporcell pipeline with the –skip_remap SKIP_REMAP and–common_variants ${VCF}. Full scripts used for souporcell processes are included below. Summary of souporcell run details can be found in Fig S12.

```
export

SINGULARITY_BIND = "${DemuxSoupDir},${MappingAnalysisDir},${FastaDir}"
singularity exec ${DemuxSoupDir}Demuxafy.sif renamer.py –bam $BAM –barcodes $BARCODES –out ${OutputName}.fq

singularity exec ${DemuxSoupDir}Demuxafy.sif minimap2 -ax splice -I 9 G -t 20 -G50k -k 11 -K 50M -w 15 –sr -A2 -B8 -O12,32 -E2,1 -r200 -p.5 -N20 -f1000,5000 -n2 -m20 -s40 -g2000 -2K50m –secondary = no ${FASTA} ${OutputName}.fq > minimap.sam

singularity exec ${DemuxSoupDir}Demuxafy.sif retag.py –sam minimap.sam –out minimap_tagged.bam

singularity exec ${DemuxSoupDir}Demuxafy.sif samtools sort minimap_tagged.bam > minimap_tagged_sorted.bam

singularity exec ${DemuxSoupDir}Demuxafy.sif samtools index minimap_tagged_sorted.bam
```

*#freebayes run:*
```
singularity exec ${DemuxSoupDir}Demuxafy.sif freebayes -f ${FASTA} -iXu -C 2 -q 20 -n 3 -E 1 -m 30 –min-coverage 6 minimap_tagged_sorted.bam > free.vcf
```

*#souporcell run.*
```
VCF = ${CurrentAnalysisDir}free.vcf

singularity exec ${DemuxSoupDir}Demuxafy.sif souporcell_pipeline.py -i ${CurrentAnalysisDir}minimap_tagged_sorted.bam -b ${BARCODES} -f ${FASTA} -t 20 -o ${OutputName} -k $N –skip_remap SKIP_REMAP –common_variants ${VCF}
```

### Pooled Pleurodeles and Notophthalmus mapping and SNP-based demultiplexing

A dual species Cell Ranger 7.0.0 reference was made using the SuperTranscriptomes and corresponding gtf files (described above) from the two species using Cell Ranger mkref command. The two species, four animal, pooled scRNA-seq from *Pleurodeles* and *Notophthalmus* dataset was then mapped to this dual species index using Cell Ranger 7.0.0 count command. In addition, the Cell Ranger 7.0.0 multi-command was used to assess multiplexing Cellplex information for all cells in the same dataset. For the Cell Ranger multi-command, the following flags were used: (expect cells 10,000, min-assignment-confidence 0.6). Souporcell demultiplexing was run identically to above on the *Pleurodeles* only samples but with N = 4, and the relevant FASTQ and dual species reference transcriptome FASTA files.

### Inbred mouse mapping and SNP-based demultiplexing

C57BL/6 data were downloaded from SRR15502048, SRR15502052, and SRR15502056 (Crowl et al, 2022). For DBA data (Shinozaki et al, 2022) possorted.bam files were downloaded from SRA. wget https://sra-pub-src-1.s3.amazonaws.com/SRR20079758/WT3_possorted_genome_bam.bam.1

wget https://sra-pub-src-1.s3.amazonaws.com/SRR20079759/WT2_possorted_genome_bam.bam.1
wget https://sra-pub-src-2.s3.amazonaws.com/SRR20079760/WT1_possorted_genome_bam.bam.1.

bamtofastq v1.4.1 was used to generate FASTQ files for subsequent mapping. A Cell Ranger reference was obtained from the 10x Genomics website (https://cf.10xgenomics.com/supp/cell-exp/refdata-gex-mm10-2020-A.tar.gz) and Cell Ranger 7.0.0 count command was then used to map and count reads from each strain and dataset.

A VCF file and index were obtained from https://ftp.ebi.ac.uk/pub/databases/mousegenomes/REL-2112-v8-SNPs_Indels/mgp_REL2021_snps.vcf.gz, https://ftp.ebi.ac.uk/pub/databases/mousegenomes/REL-2112-v8-SNPs_Indels/mgp_REL2021_snps.vcf.gz.csi, and chromosomes were renamed as previously described in the zebrafish and green monkey sections above.

bcftools annotate –rename-chrs Conv.chr mgp_REL2021_snps.vcf.gz | bgzip > rename.mgp_REL2021_snps.vcf.gz

To pool bams, we modified the original synth_pool.py script from Vireo's GitHub repository which was memory intensive and only allowed for pooling in the presence of a VCF file (https://zenodo.org/record/7929057). To enable more widespread use, we now stabilized memory use throughout the pooling and added an option (–noregionFile) which can pool in the absence of a VCF file. This means that species that do not possess a VCF file can still do the in silico ground truth benchmarking we performed in this study. A pull request has been initiated to propagate the changes to Vireo's main GitHub repository (https://github.com/single-cell-genetics/vireo/pull/81). The modified version was used to pool the mouse data below. Note: if the pull request is accepted then calling the synth_pool.py script from Vireo's GitHub repository will in fact be this modified script with new options added. This script can produce identical pooled BAM files to the original.

python synth_pool.py -s possorted_genome_bam.bam,possorted_genome_bam.bam,possorted_genome_bam.bam -b barcodes.tsv,barcodes.tsv,barcodes.tsv -d 0.1 -o mixed_mice -p 6 -r rename.mgp_REL2021_snps.vcf.gz –randomSEED 50.

Vireo and souporcell were run as per the previous species and assignment percentages for souporcell were assessed via awk -F "\t" "{print $2}" clusters.tsv | sort | uniq -c | sort -nr followed by calculation (singlet assignments/total cells)*100. For Vireo, the "final donor size" numbers were used and percentage assigned calculated by ((donor0+donor1+donor2)/unassigned)*100.

## Souporcell analysis details summary

A summary figure reviews the computational details used to run souporcell on the above datasets (Fig S12). Overall, the standard souporcell default pipeline was used initially, but additional flags or pieces of the souporcell pipeline were run externally when this failed because of memory constraints or other problems. All SNP demuxers were run from a Demuxafy (Neavin et al, 2022 Preprint) singularity container (image version 1.0.3). SNP numbers in each VCF were counted using: grep "##" VCFname | wc -l.

For analyses of *Pleurodeles* and dual species datasets, the first two steps of the souporcell pipeline were run separately and then the output from these was introduced back into the souporcell pipeline for completion (Fig S12). This allowed us to adjust computational parameters in the remapping stage that permitted the function to finish.

## Analysis and benchmarking of souporcell assignments: R analysis

A two part analysis in R and then Python was used to evaluate the efficacy of souporcell demultiplexing for each dataset. Scripts in R (version 4.1.2) primarily using Seurat (version 4.1.0) (Hao et al, 2021) were used to evaluate souporcell cell assignments found in the clusters.tsv file through comparison with known cell identities or cell assignments based on fluorescence, barnyard analysis, or CMO labeling. Full R scripts for all analyses are deposited in the GitHub page for clarity (https://github.com/RegenImm-Lab/SNPdemuxPaper). Seurat was used to import and analyze single-cell gene expression data for all datasets and to analyze the multiplexing capture data for the dual-species Cellplex (CMO)-labeled dataset.

## Cell filtering

For UMAP plots and all bar plots in benchmarking analysis, cells for all experimentally pooled datasets were filtered before analysis to select for cells most likely to have accurate calls by the respective benchmarking assay. Cell Ranger default filtered cells were parsed to select cells with between 5,000 and 40,000 mapped reads for each dataset. In addition, for Figs 2 and 3, we removed cells lacking counts of any of the three fluorescent mRNAs. Cell assignments based on fluorescence or Cellplex labeling were made through the analysis of fluorescent reads or Cellplex CMO reads by the Seurat MULTIseqDemux function (autoThresh = T) (McGinnis et al, 2019). Although the MULTIseqDemux function is written to assign cell identities based on CMO labels, we found that it works well with data from overexpressed fluorescent gene mRNAs. For analysis purposes, souporcell demultiplexing numerical cell labels were adjusted to relevant sample names based on correlation analyses to each relevant benchmarking demultiplexing method. UMAP (McInnes et al, 2018 Preprint) and upset plots (Lex et al, 2014) were generated in R scripts and annotated in Affinity Designer. A dataframe with compiled cell assignment information was exported to Python for further analysis.

Session info including package numbers for R analyses are embedded in the GitHub page (https://github.com/RegenImm-Lab/SNPdemuxPaper), and included R version 4.1.2 (2021-11-01).

## Analysis and benchmarking of souporcell assignments: Python analysis

Analyses of read depth versus Cell ID% and quantitative benchmarking of souporcell demultiplexing results was carried out in Google Colab notebooks shared below, using Python version 3.7, NumPy version 1.21.6 (Harris et al, 2020), Pandas version 1.3.5 (Reback et al, 2021), Matplotlib version 3.2.2 (Hunter, 2007), SciPy version 1.7.3 (Virtanen et al, 2020), Seaborn version 0.11.2 (Waskom, 2021).

Google Colab links for analyses are located here:

*Xenopus*: https://colab.research.google.com/drive/1lO4ny8Uv9n1lPIbHmZKFPyxWl7gjmZPs?usp=sharing.

*Pleurodeles*:

https://colab.research.google.com/drive/1Zbbpi1WwfKGwrrFhuecrHE3lSjP0Pzsz?usp=sharing.

*Pleurodeles* and *Notophthalmus* cellplex:

https://colab.research.google.com/drive/12ZNvvfiUt3DL6UpTg8BjQMSd4Y-6yM3u?usp=sharing.

*Pleurodeles* and *Notophthalmus* barnyard:

https://colab.research.google.com/drive/1JS8kRUGAioDM2IvYa4oBDzNMKLxBhHV_?usp=sharing.

Zebrafish, axolotl, and green monkey:

https://colab.research.google.com/drive/1yXzE3WJ05hEJKdy7owiXOpCjYvUm4DDJ?usp=sharing.

### Calculation of cell ID %

All cells were sorted by total mapped read depth, and binned into 40 groups before calculating Cell ID%, and average total and fluorescent read depth were relevant. Cell ID % is defined as the percentage of cells in each bin that were assigned an individual sample identity (non-doublet, non-negative result) by a demultiplexing method. Cell ID value for each bin was then plotted against average total mapped reads. For datasets including fluorescent transgenic lines, binned data are also colored by the average number of summed mapped fluorescent reads per cell.

Bar plots: filtered datasets were subset by the animal or animal group assignment from each demultiplex method being used to benchmark souporcell results. Within those subsets, the total cell quantity of cells assigned to each identity by souporcell was plotted (left plots). Alternatively, within each benchmarking demux result subset, the percentage of cells assigned to each identity by souporcell was calculated by dividing by total cells assigned to that identity by the benchmarking demuxer and multiplied by 100.

### SNP density calculations

VCF files were generated from 10x BAMs and the species-specific reference using VCFtools v1.11 with:

bcftools mpileup -f $GENOME -b bamlist –threads 10 | bcftools call -m -Oz -f GQ –threads 10 -o allsites.vcf

SNP density per kilobase was then calculated using VCFtools v 0.1.15:

vcftools –SNPdensity 1000 –gzvcf allsites.vcf.gz

This outputs an out.snpden file, and average SNP density across all sites was calculated using: awk "{ total + = $4; count++ } END { print total/count }" out.snpden

## Data Availability

Data are available on ArrayExpress with accession E-MTAB-12186 for three animal pooled *Pleurodeles* splenocyte scRNA-seq and ArrayExpress accession E-MTAB-12182 for four animal pooled *Pleurodeles* and *Notophthalmus* splenocyte scRNA-seq. Code used

to analyze the data are present in the Materials and Methods section, in linked Colab notebooks, or via GitHub (https://github.com/RegenImm-Lab/SNPdemuxPaper) All other data used in the study were from previously published works of which accessions are noted in the Materials and Methods section.

## Supplementary Information

## Acknowledgements

We thank the Eukaryotic Single Cell Genomics core at SciLifeLab for the generation of 10x Genomics libraries of *Pleurodeles* and *Notophthalmus* samples. Sequencing was performed at NGI Genome Center which is funded by RFI/VR and Science for Life Laboratory, Sweden. The computations, storage, and data handling were enabled by resources provided by LUNARC which is part of the Swedish National Infrastructure for Computing (SNIC) and is partially funded by the Swedish Research Council through grant agreement no. 2018-05973. We thank Tomás Pires de Carvalho Gomes and Ashley Maynard for graciously sharing their data with us early and Tomás Pires de Carvalho Gomes for providing feedback on this study. We also thank Garrett Dunlap for critical review and feedback. In addition, we thank the Simon Laboratory for insightful discussions and help with animal care. Diagrams with animals were generated with Biorender.com. ND Leigh receives funding from the Knut and Alice Wallenberg Foundation and the Swedish Research Council (Registration # 2020-01486). A Simon receives funding from European Research Council, Swedish Research Council, and Cancerfonden.

### Author Contributions

JF Cardiello: conceptualization, data curation, formal analysis, validation, investigation, visualization, methodology, and writing—original draft, review, and editing.
A Joven Araus and S Giatrellis: investigation, methodology, and writing—review and editing.
C Helsens: validation, investigation, methodology, and writing—review and editing.
A Simon: resources, project administration, and writing—review and editing.
ND Leigh: conceptualization, resources, data curation, formal analysis, supervision, funding acquisition, validation, investigation, visualization, methodology, and writing—original draft, review and editing.

### Conflict of Interest Statement

The authors declare that they have no conflict of interest.

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
