## [Reviewer comments · Life Science Alliance]

Life Science Alliance

Evaluation of genetic demultiplexing of single cell sequencing data from model species

Joseph Cardiello, Alberto Joven Araus, Sarantis Giatrellis, Clement Helsens, Andras Simon, and Nicholas Leigh
DOI: <https://doi.org/10.26508/lsa.202301979>

Corresponding author(s): Nicholas Leigh, Lund University

Review Timeline:

Submission Date:	2023-02-08
Editorial Decision:	2023-02-13
Revision Received:	2023-04-06
Editorial Decision:	2023-04-27
Revision Received:	2023-05-03
Accepted:	2023-05-03

Transaction Report:

Please note that the manuscript was reviewed at Review Commons and these reports were taken into account in the decision-making process at Life Science Alliance.

February 13, 2023

Re: Life Science Alliance manuscript #LSA-2023-01979

Dr. Nicholas Leigh
Lund University
Laboratory Medicine
Sölvegatan 17
Lund 22184
Sweden

Dear Dr. Leigh,

Thank you for submitting your manuscript entitled "Accurate genotype-based demultiplexing of single cell RNA sequencing samples from non-human animals" to Life Science Alliance. We invite you to re-submit the manuscript, revised according to your Revision Plan.

Thank you for this interesting contribution to Life Science Alliance. We are looking forward to receiving your revised manuscript.

Sincerely,

B. MANUSCRIPT ORGANIZATION AND FORMATTING:

Reviewer #1 (Evidence, reproducibility and clarity (Required)):

“Cardiello et al tested if souporecell (<https://pubmed.ncbi.nlm.nih.gov/32366989/>) can be used to demultiplex samples for some model organisms, based on identified SNPs. For this, they used synthetic multiplexed data, publicly available datasets and some new datasets, spanning samples from five model organisms. Their analysis indicates that souporecell could be used to demultiplex scRNA-seq experiments for multiple species, which offers a cost-beneficent approach.

The manuscript reads well and shows this approach can work for different model organisms. However, unfortunately, I am confused about the amount of novelty in this manuscript. The method, souporecell, is already published. The authors indicate souporecell is not validated in non-human samples, but the original paper states that their method works with malaria parasite data (Fig 3b, FigS4). Adapting and using an available tool for different model organisms is good and groups working on different model organisms may find this manuscript useful, but the same could be said for the original article. Due to these reasons, I am not sure whether this manuscript has novelty sufficient for publication.”

Our response:

This comment helped us realize that our novelty was not clearly stated in the first version of the manuscript. We revamped our Introduction, Results, and Discussion to highlight the novelty of this work more clearly. We also clarified what has been previously performed and how this current manuscript provides novel insight into multiple previously unanswered questions which broadly extend the utility of SNP-based demultiplexing. In general, as biological sciences become more computational, benchmarking tools developed for other systems/species is essential. We thus also included analysis with more SNP-based demultiplexers and provided a decision tree (Supplemental Figure 2) to help readers to understand that the results from souporecell and multiple other SNP-based demultiplexers were very similar in our hands. This decision tree explains why we then chose to move forward with a more extensive analysis of results from souporecell.

We clarified our verbiage regarding the definition of “validation”. We define validation as establishing the accuracy or validity of a method. Therefore, validation of SNP-based demultiplexing for use in non-human species requires comparison to an already proven, orthogonal method, such as a wet-lab based demultiplexing approach. The original souporecell paper does not validate (i.e., confirm with an orthogonal wet-lab method) the results from souporecell. The fact that SNP-based demultiplexing is not being used nor tested in non-human organisms made it unclear how and if these approaches would work in other species. Human samples are expected to perform exceptionally well in this approach due to their extremely high genetic diversity and wealth of available genomic resources. Thus, while it was encouraging that the original souporecell authors chose to apply their algorithm to a non-human (e.g., malarial parasite) dataset, this analysis was not validated by ground truths arrived from any technology other than SNP-based demultiplexers.

(Reviewer #1): *I also wrote down two minor points below:*

“1- Doublets assigned by souporecell compared to the fluor-based assignment look random. In Fig 2 doublet recovery rate looks smaller, and in fig 3 doublet rate prediction looks more random. This is a bit confusing. Is there any explanation for this?”

Our response: We have expanded more in the paper on doublet detection and the drawbacks of SNP-based demultiplexing methods for doublet detection. We also used other SNP-based demultiplexers on both zebrafish and African Green Monkey data and found that all SNP-based demuxers, minus scSplit, made similar mistakes. This suggests that missed doublet assignments are not random. While the fluorescent-based experiments are informative in a broad sense, the very low counts of fluorescent genes makes the doublet assignment by this specific approach very difficult and likely unreliable. This is why we also took other approaches. In general, our data indicate that SNP-based demultiplexers are inadequate as standalone doublet detectors. Analysis in this revised manuscript (Supplemental Figure 4) suggests that the number of animals in a sample may contribute to poor performance and increasing from three to five animals improves doublet detection. To combat this, in the Discussion, we suggest that moving forward with a combination of doublet detection methods is best practice when working with large cell numbers which will lead to high doublet numbers. We highlight some tools that may be useful for future users.

(Reviewer #1): *“2- The authors discussed the immune system cells might show some variability in their discussion (referring to fig 3), but this is not clearly shown in the figures as data. Having a percentage bar graph could make it clearer for the readers.”*

Our response: Thanks for pointing this out. We have now broken up UMAPs by sample in Supplemental Figure 8 and included an informative table there too. Together, these clearly highlight the between sample heterogeneity.

From Reviewer 1:

“More generally, showing more direct evidence for the variability of different cell types (not just the immune system) could be informative for scRNA-seq users.”

Our response: We appreciate the comment on more clearly informing scRNAseq users of the potential for between sample variation. We have now thoroughly cited papers speaking to this point and made them clearer in our manuscript. We believe an extensive analysis of other published single cell datasets in which we attempt to identify artifactual results and animal-to-animal variability would be harmful to future open science efforts. Past papers have extensively demonstrated the issue of batch effects and animal-to-animal variation in scRNA-seq datasets, and the requirement for biological replicates to facilitate differential expression analysis.

Reviewer #2 (Evidence, reproducibility and clarity (Required)):

Major comments:

*“1. SNP-based demultiplexing performed well on some species, such as zebrafish and Africa green monkey, from which over 90% of the cells analyzed were correctly identified. However, this accuracy decreases in *Pleurodeles* samples when a common SNPs VCF file is absent (Fig. 3). It showed that cell identity can be more precisely defined with the increase of average read depth (Fig 3B). So, I am wondering whether the mis-defined cells shown in Fig. 3E, actually are cells with lower reads. It is better if the authors can test such a correlation between the cell identity and the depth of reads using the data from Fig. 3E.”*

Our response: We are thankful to reviewer #2 for raising such a great point. As reviewer #2 suggests, we do see the accuracy of the benchmarking results for this experiment increase with increasing sequence depth/cell quality. We demonstrate this more directly with a new figure, plotting percent agreement between the two demultiplexing approaches against average cell read depth (Supplemental Figure 8I). However, the reasons for the influence of read depth on this benchmarking accuracy are potentially more complex than just higher accuracy of souporecell in higher quality cells: The fluorescent-based demultiplexing that is being used for “ground truth” in benchmarking souporecell for this specific figure is more accurate in cells with higher read depth because more fluorescent gene reads are likely to be captured (as seen by the color of dots in Figure 3B). Therefore analyzing the accuracy of souporecell relative to fluorescent-based demultiplexing over varying read depths can be confusing because both methods are likely to improve in accuracy with higher read depth. Figure 3B was originally included in this paper to illustrate this concern, and to demonstrate why we chose to benchmark only the cells with sufficient read depth (read depth between 5K, and 40K, and >1 fluorescent gene read per cell).

Nevertheless, to address this question about a relationship between the accuracy of souporecell cell assignments and average cell read depth, we made a similar plot to the new Supplemental Figure 8I, but with our dataset that included the cellplex labels (i.e., Figure 4 data). Here, our ground truth approach used for benchmarking souporecell was cellplex and should be independent from the gene expression read depth because the cellplex reads are a distinctly sequenced library (Supplemental Figure 10). In this new figure we do once again see a relationship between read depth and souporecell benchmarking accuracy, suggesting that the accuracy of cell assignments from SNP-based demultiplexing do appear to be influenced by read depth, as suggested by reviewer #2. These new figures help to clarify this important point.

We also performed a test with and without a common VCF file for the zebrafish data to see if the absence of a common SNP file was a potential reason for the *Pleurodeles* analysis decreased assignments. We found that less than 1% of cells were assigned differently with or without the common SNPs VCF in the zebrafish dataset and added this information to the manuscript. We also added comments on our experiences with and without common SNPs VCFs in the methods as we did see some issues with the use of common SNPs VCF for Freemuxlet. We expect that the human common SNPs VCF is of far higher quality than any

other species and thus generating VCF from the data itself may be a safer approach for most species.

(Reviewer #2)

“2. Please discuss limitations of this approach in the manuscript. (1) To which extent, when SNPs are roughly present in the individuals of same species, SNP-based demultiplexing can be applied, e.g., individuals from an inbred strain (c57bl6 mice) would not work.(2) The authors experimentally tested two newt species using SNP-based demultiplexing. When multiple species are experimentally applied, may the cell/nuclei size variation cause problem?”

Our response:

We agree it is critical to define the limits of SNP-based demultiplexing. We have added data and comments to clarify these limits. Regarding data, we have provided information on SNP density of the animals in this study (see section Identifying the limits of SNP-based demultiplexing). This provides a reference level of variation for those working in other species as to whether SNP-based demultiplexing may be a viable approach. We also confirmed SNP-based demultiplexing within inbred mouse strains is unsuccessful (see section, Identifying the limits of SNP-based demultiplexing).

Lastly, given the wide array of species that SNP-based demultiplexing could potentially be applied to, we also provide scripts and examples that will facilitate our *in silico* pooling analysis on any species of interest. Testing directly whether SNP-based demultiplexing works in each individual use case will provide assurances that it can be used without orthogonal labeling methods. We highlight this in the Discussion. We also added to the Discussion a comment on cell/nuclei size as this is a good point that users should consider before embarking on multi-species mixing experiments.

(Reviewer #2)

“3. What is the upper limit number of samples when using this model. Please make some estimation or discussion about it.”

Our response: This is a pressing question for the future of SNP-based demuxing and it deserves further discussion in this manuscript. Regarding souporecell the authors suggest (github thread) an upper limit in human samples (worked on 21 human samples, may work in up to 40). We have added this information and link to the Discussion in the revised manuscript. At this point, we have no reason to believe that the limit on sample numbers should be different in other species with similar levels of genetic variation.

We also added an analysis of SNP-based demultiplexing of a pool of 30 zebrafish embryos. We also provide warnings about interpreting such results and the limitations of demultiplexing large pools in the Discussion. We further elaborate on a workaround of layered multiplexing (i.e., pairing SNP-based demultiplexing with one of the wet-lab based approaches

such as cell hashing). With the methods provided in this manuscript a lab interested in investigating this limit in their model organism of choice could do so.

(Reviewer #2) Minor comments:

“1. Please add an algorithm principle of this model.”

Our response: Thanks for this suggestion. We clearly cite all the SNP demuxing tools that we used in the paper and since we did not develop these tools ask the readers to go there for more detailed information about the algorithms. We instead provide a decision tree in Supplemental Figure 2 to aid readers in selecting the tools that can be used on their datasets given the varying input requirements. We have also made clear any changes we made to the tools defaults in Supplemental Table 3.

(Reviewer #2)

“2. Give a clear definition of doublets including the ground truth and Souporecell result.”

Our response: Thanks for the suggestion. We have added a clear definition of doublets to the manuscript and provide informative names based on doublets from ground truth assignments or souporecell assignments.

(Reviewer #2)

“3. Authors should indicate the time cost of running one round of such analysis, the minimal computational requirements?”

Our response: This is an important point and that will be helpful to readers. We have provided a comparison between tools of time costs and memory usage in Supplemental Table 1. This is a small dataset but illustrates potential differences between tools. We also provide this same information for a “real” souporecell run on a new large dataset (zebrafish 30 embryo pool) we included in the paper.

Reviewer #3 Major comments:

“The manuscript makes a convincing case for the ability of a preexisting SNP-based demultiplexing tool, called souporecell, to demultiplex pooled samples. The study uses three methods for validation: 1. In silico data pooling; 2. Pooling of transgenic lines; 3. Pooling of cells tagged with CMOs (10x genomics). The results are consistent across experiments.

The authors propose that souporecell is a solution for demultiplexing pooled samples whenever sample tagging methods are not feasible. Although the authors test this approach in several species and conditions, the validation does not cover all possible cases and situations, obviously. Indeed, the authors recommend potential users to run pilot validation experiments with a secondary demultiplexing methods.

However, the manuscript would become more useful if the following points are addressed:

First, what is the genetic relatedness of the individuals pooled in the experiments? What is the SNP frequency in the samples analyzed, and how does that compare to SNP frequency in mouse strains? (The number of SNPs in the VCF is reported in a supplementary table but not discussed in the main text). This point is extremely important: as the authors mention, it is not possible to demultiplex samples from the same mouse strain. Inbreeding is relatively common in laboratory species, even unconventional ones; therefore, information on genetic relatedness and SNP rate would help readers assess whether SNP-based demultiplexing has a good chance to work in their systems. Addressing this point does not require any additional experiments, and computing from the single-cell reads how many SNPs distinguish the individuals pooled here should be straightforward.”

Our response: We appreciate the comments raised by reviewer #3. These are valuable critiques and have greatly improved the utility of the manuscript. Regarding genetic relatedness, please see response to reviewer #2 above. We now provide information about genetic diversity of the different species of animals used in this study. This is critical new information that greatly increases the impact of this manuscript and will provide a litmus test for other labs to gauge if SNP-based demultiplexing may benefit them.

(Reviewer #3):

“Moreover, the relatively limited number of samples pooled does not validate the use of souporecell with a larger number of samples. For example: in developmental studies, often dozens of embryos are collected and pooled. What are the potential caveats of using souporecell for demultiplexing larger number of samples? The Discussion would be a good place to warn potential users of the limitations of the approach.”

Our response: We agree this could still be a limitation, and for developmental studies with multiple dozens of samples, further exploration of optimal demultiplexing methods or the combination of computational and wet lab-based demux methods is required. We further comment on the limits in response to reviewer #2 above. Importantly, this paper provides a thorough framework for labs with their model organism of choice to explore this question.

(Reviewer #3) Minor comments:

“- is the accuracy of doublet detection rate a function of number of samples? This can be tested by repeating the monkey in silico experiment with three individuals.”

Our response: This is a good point. As suggested, we tested this with the monkey dataset and found that indeed it appears sample number impacts doublet detection. This is a critical new addition to the manuscript and one that all labs pooling samples must consider in optimizing experiments. We also added some notes on this in the Discussion as practically speaking, the

more samples in the pool the higher the number of heterotypic doublets (i.e., cells from two individuals) should be detected. As pool size increases it is possible to detect more heterotypic doublets because homotypic doublets (i.e., two cells from the same individual) would become an increasingly smaller portion of doublets. For pools of equal cell number the expectation is that $1/N$ of all doublets are homotypic (<http://bioconductor.org/books/3.13/OSCA.advanced/doublet-detection.html#doublet-detection-in-multiplexed-experiments>). This is another advantage of sample pooling that we did not clearly articulate in the first version of the manuscript.

April 27, 2023

RE: Life Science Alliance Manuscript #LSA-2023-01979R

Dr. Nicholas Leigh
Lund University
Laboratory Medicine
Sölvegatan 17
Lund 22184
Sweden

Dear Dr. Leigh,

Thank you for submitting your revised manuscript entitled "Evaluation of genetic demultiplexing of single cell sequencing data from model species". We would be happy to publish your paper in Life Science Alliance pending final revisions necessary to meet our formatting guidelines.

- please add a separate conflict of interest statement to your main manuscript text
- please add the supplementary figure legends to the main manuscript text
- please add a figure callout for Figure 1E-F to the main manuscript text
- please add the panels A and B to your Figure S1 in the figure itself and add a figure callout for Figure S1A to your main manuscript text

Figure Check:

- please add scale bars to all of the panels in Figure S13

A. FINAL FILES:

B. MANUSCRIPT ORGANIZATION AND FORMATTING:

Sincerely,

Reviewer #1 (Comments to the Authors (Required)):

I express my gratitude to the authors for their comprehensive revision, which includes the title and highlights the novelty of their research. The work was already commendable before the revision, and the revised version effectively showcases its originality, and is more accessible to a broader audience. The results of this study will prove invaluable to the various communities utilizing single-cell transcriptomics. I wholeheartedly endorse its publication with this version.

Reviewer #2 (Comments to the Authors (Required)):

Dear editor,

the authors have addressed all my concerns in the revised manuscript. I believe now it is ready to be published at present format.

Reviewer #3 (Comments to the Authors (Required)):

In this revised version, Cardiello and colleagues addressed thoroughly all the main points raised during the previous peer-review cycle.

In particular, the comparison of different demultiplexing methods presented in the supplementary figures, and the decision tree in supply. fig. 2 are going to be extremely valuable for other researchers in the community.

I recommend the publication of this article in Life Science Alliance

May 3, 2023

RE: Life Science Alliance Manuscript #LSA-2023-01979RR

Dr. Nicholas Leigh
Lund University
Laboratory Medicine
Sölvegatan 17
Lund 22184
Sweden

Dear Dr. Leigh,

Thank you for submitting your Methods entitled "Evaluation of genetic demultiplexing of single cell sequencing data from model species". It is a pleasure to let you know that your manuscript is now accepted for publication in Life Science Alliance. Congratulations on this interesting work.

DISTRIBUTION OF MATERIALS:

Again, congratulations on a very nice paper. I hope you found the review process to be constructive and are pleased with how the manuscript was handled editorially. We look forward to future exciting submissions from your lab.

Sincerely,
